# An innovative approach using CRISPR-ribonucleoprotein packaged in virus-like particles to generate genetically engineered mouse models

Tae Yeong Jeong [1,2,3,13], Da Eun Yoon [1,2,3,4,13], Sol Pin Kim[5,6,13], Jiyun Yang [1,2,3,13], Soo-Yeon Lim[5], Sungjin Ok[1], Sungjin Ju [1,2,3], Jeongeun Park[1,3,5], Su Bin Lee[5], Soo-Ji Park [1,2,3], Sanghun Kim[2,7], Hyunji Lee[2,7], Daekee Lee [8], Soo Kyung Kang [5,6], Seung Eun Lee[9], Hyeon Soo Kim[10], Je Kyung Seong [5,6,11,12] ✉ & Kyoungmi Kim [1,2,3,5] ✉

Genetically engineered mouse models (GEMMs) are crucial for investigating disease mechanisms, developing therapeutic strategies, and advancing fundamental biological research. While CRISPR gene editing has greatly facilitated the creation of these models, existing techniques still present technical challenges and efficiency limitations. Here, we establish a CRISPR-VLP-induced targeted mutagenesis (CRISPR-VIM) strategy, enabling precise genome editing by co-culturing zygotes with virus-like particle (VLP)-delivered gene editing ribonucleoproteins (RNPs) without requiring physical manipulation or causing cellular damage. We generate *Plin1*- and *Tyr*-knockout mice through VLP-based SpCas9 or adenine base editor (ABE)/sgRNA RNPs and characterize their phenotype and germline transmission. Additionally, we demonstrate cytosine base editor (CBE)/sgRNA-based C-to-T substitution or SpCas9/sgRNA-based knock-in using VLPs. This method further simplifies and accelerates GEMM generation without specialized techniques or equipment. Consequently, the CRISPR-VIM method can facilitate mouse modeling and be applied in various research fields.

Genetically engineered mouse models (GEMMs) are widely used in studies of gene functions, biomedical mechanisms, and the causes of disease, as well as basic research[1,2]. GEMMs that mimic human genetic mutations in mouse models are essential for verifying drug efficacy and applying potential treatments. Among the available gene-editing technologies, clustered regularly interspaced short palindromic repeats (CRISPR), a third-generation tool, is widely used to produce GEMMs due to its high editing efficiency and ease of use compared to earlier methods[3].

Advancements in the CRISPR system have led to the development of base editors, such as the cytosine base editor (CBE)[4] and adenine base editor (ABE)[5], which enable precise nucleotide conversion. In addition, the development of prime editor (PE)[6], which is capable of performing small insertions, deletions, and base substitutions, has expanded the possibilities for genome editing. These innovations significantly enhance the versatility and precision of GEMM generation.

Traditionally, 1-cell stage fertilized eggs, also known as zygotes, are widely used for animal model generation. Currently, available methods involve direct microinjection or electroporation of plasmids, mRNA, or ribonucleoprotein (RNP) of the CRISPR system into zygotes using a microinjector or electroporator for introducing desired

mutations in mice[3,7–10]. For gene editing of zygotes, RNP is considered safer than plasmids or mRNAs due to reduced off-target effects[11–16]. However, the applications of RNPs are limited owing to difficulties in protein purification across various CRISPR systems. In addition, physical delivery methods are challenging because they require specialized techniques and expensive equipment. Furthermore, these methods can potentially damage embryos[17,18].

Virus-like particles (VLPs) are currently utilized as vaccines and drug delivery vehicles, taking advantage of their structural similarities to viruses while lacking viral genetic material[19]. Owing to these characteristics, VLPs have attracted considerable attention as pioneering carriers that ensure the safe and efficient delivery of gene editing tools in advanced biomedical applications[20–29]. Research on murine leukemia virus (MLV)-derived VLPs based on the CRISPR/Cas9, ABE, and PE system demonstrated efficient target gene editing by delivering VLPs into cells and mice[30–32]. A study on GEMM production using VLPs reported the microinjection of MLV-like particles loaded with Cas9-sgRNA RNPs (nanoblades) into the perivitelline space of mouse zygotes[27]. However, this method remained limited for GEMM production because the microinjection technique is still required. To address these challenges, we propose an innovative and convenient approach, the CRISPR-VLP-induced targeted mutagenesis (CRISPR-VIM) method. This method enables targeted mutagenesis by simply co-culturing zygotes with VLP-packaged CRISPR-RNP, eliminating the need for other physical delivery methods. This approach can simplify and accelerate GEMM generation, contributing to the advancement of research in various fields using animal models.

## Results

### Optimization of delivery conditions for CRISPR-RNP packaged in VLPs in cell lines and mouse zygotes

We comprehensively evaluated the gene editing efficiency of VLPs packaged with SpCas9/sgRNA or ABE8e/sgRNA targeting *TTN*, *HEK3*, and *HBB* in HEK293T and ARPE19 cells (Fig. 1a and Supplementary Fig. 1a–l). Both types of VLPs exhibited varying gene editing efficiencies depending on the target gene and cell line. However, for most targeted genes, gene editing efficiency reached saturation in groups treated with 5, 10, or 25 μl of VLPs. Remarkably, specific targets showed high editing efficiency of up to 97.3% even with 1 μl VLPs. Subsequently, we expanded our gene editing using VLP-packaged SpCas9/sgRNA and ABE8e/sgRNA to target various mouse genes in mouse embryonic stem cells (mESCs) and Neuro-2a cells (Supplementary Fig. 1m, n and Fig. 1b). The SpCas9/sgRNA packaged in VLPs achieved up to 99.8% gene editing efficiency on *Fgfr3*, *Gata3*, *Plin1*, and *Tyr* targets (Fig. 1b). ABE8e/sgRNA packaged in VLPs showed high A-to-G conversion frequency of up to 98.4% in *Plin1*, *Dnmt1*, *Gata3*, and *Kcnq4* targets (Fig. 1c). These results demonstrate that the VLP-packaged CRISPR-RNP system can achieve high gene editing efficiency across various targets and cell types.

Previous studies used microinjection methods to deliver VLPs directly into the perivitelline space of a mouse zygote[27]. However, recent studies have shown that the mammalian zona pellucida permits the passage of 20–200 nm viruses and nanoparticles up to 250 nm in size[33–35]. Therefore, we postulated that 100–150 nm VLPs could pass through the zona pellucida. Thus, we conducted experiments applying VLPs, including several CRISPR systems to mouse zygotes. First, we tested whether VLP-packaged CRISPR-RNP could induce targeted mutagenesis in mouse zygotes by their co-culturing in the medium and yielded positive results. Notably, SpCas9/sgRNA delivered via VLPs achieved gene editing efficiencies up to 99.9% in embryos (Fig. 1d–i, Supplementary Fig. 2, and Supplementary Fig. 6a–e).

We co-cultured VLP-packaged ABE8e/sgRNA targeting *Tyr* or *Plin1* at a concentration of 10% of the total medium volume for varying treatment times (0, 1, 5, 10, and 20 hours) to optimize the treatment condition of CRISPR-RNP packaged in VLPs in mouse zygotes. Notably, gene editing efficiencies of up to 49.4% for *Tyr* and 76.5% for *Plin1* were observed at 20 h of VLP-packaged CRISPR-RNP treatment in mouse zygotes (Fig. 1e, f and Supplementary Fig. 6a, b).

Afterward, we conducted experiments to determine the optimized treatment volume to achieve high gene editing efficiency and low cytotoxicity in mouse embryos. VLP-packaged ABE8e/sgRNA was delivered to mouse zygotes at 10% or 20% of the total culture medium volume for 20 h. The 20% treatment group of VLP-packaged ABE8e/sgRNA exhibited significantly higher gene editing efficiency for all targets (*Plin1*, *Dnmt1*, and *Gata3*) compared to the 10% treatment group (Fig. 1g–i and Supplementary Fig. 6c–e). Specifically, *Plin1*, *Dnmt1*, and *Gata3* achieved A-to-G conversion efficiencies of 76.1%, 49.6%, and 99.4%, respectively, in the 20% treatment group of VLP-packaged ABE8e/sgRNA (Fig. 1g–i). We also evaluated the development rates of morula and blastocyst stage 4 days after treatment with 10% or 20% VLPs to evaluate VLP cytotoxicity. Development rates had no significant differences in both 10% and 20% treatment groups compared to the untreated group (Fig. 1j and Supplementary Fig. 9a). Our results demonstrate that simply co-culturing mouse zygotes with VLP-packaged CRISPR-RNP for 20 hours in the medium induces targeted mutagenesis with high gene editing efficiency and no observable cytotoxicity.

### Generation of *Plin1*-knockout mouse via CRISPR-Cas9/sgRNA packaged in VLPs

We utilized the innovative CRISPR-VIM method to easily and efficiently generate *Plin1*-knockout mice. We produced VLPs carrying SpCas9/sgRNAs that target *Plin1* exon 2 and co-cultured zygotes with 10% or 20% VLPs for 20 hours in vitro (10%, $1.50 \times 10^9$ VLPs; 20%, $3.00 \times 10^9$ VLPs) (Fig. 2a). VLP-treated 2-cell stage embryos were transplanted into oviducts of surrogate mothers, resulting in 12 *Plin1*-mutant offspring in the 10% VLPs treatment group and 7 *Plin1*-mutant offspring in the 20% VLPs treatment group with no detectable off-target effects (Fig. 2b and Supplementary Figs. 3a, b, 4a, 6f). Among *Plin1*-knockout mice, #53 M mouse born in the experimental group treated with 20% of VLP-packaged SpCas9/sgRNA showed 44.7% editing efficiency and exhibited a heterozygous genotype caused by a frameshift-inducing 29-nucleotide deletion in *Plin1* exon 2 (Fig. 2c). Then, we crossed *Plin1*-mutant mouse (#53 M) with wild-type mouse to identify germline transmission and got the F1 generation in 9 out of 13 mice carrying the mutant genotype achieved by deletion of 29 nucleotides (Fig. 2d, e). Since PLIN1 localizes to the surface of lipid droplets and is abundant in the adipose tissue[36–38], we confirmed the *Plin1*-knockout phenotype in epididymal white adipose tissue (eWAT) and inguinal white adipose tissue (iWAT) of F2 male mice, which were generated by crossing two heterozygous F1 mice, using immunofluorescence and western blot analysis (Fig. 2f, g). *Plin1*-knockout mice had smaller adipocyte sizes and increased infiltration of F4/80⁺ macrophages in eWAT and iWAT than wild-type mice[36] (Fig. 2h, i). We demonstrated that mouse models can be efficiently generated using the CRISPR-VIM method. Furthermore, the generated mutant mice can be reliably transmitted to the next generation, thereby enabling phenotypic analysis.

### Targeted mutagenesis in mice using CRISPR-ABE8e/sgRNA packaged in VLPs during in vitro fertilization

In vitro fertilization (IVF) method is well-established and widely used in mouse research. Specifically, IVF offers the advantage of producing a larger number of fertilized embryos compared to natural mating after superovulation. Given these advantages, we aimed to integrate the CRISPR-VIM method into the IVF process, which could enhance the efficiency of generating GEMMs. We applied VLP-packaged ABE8e/sgRNA targeting *Gata3* or *Plin1* at different

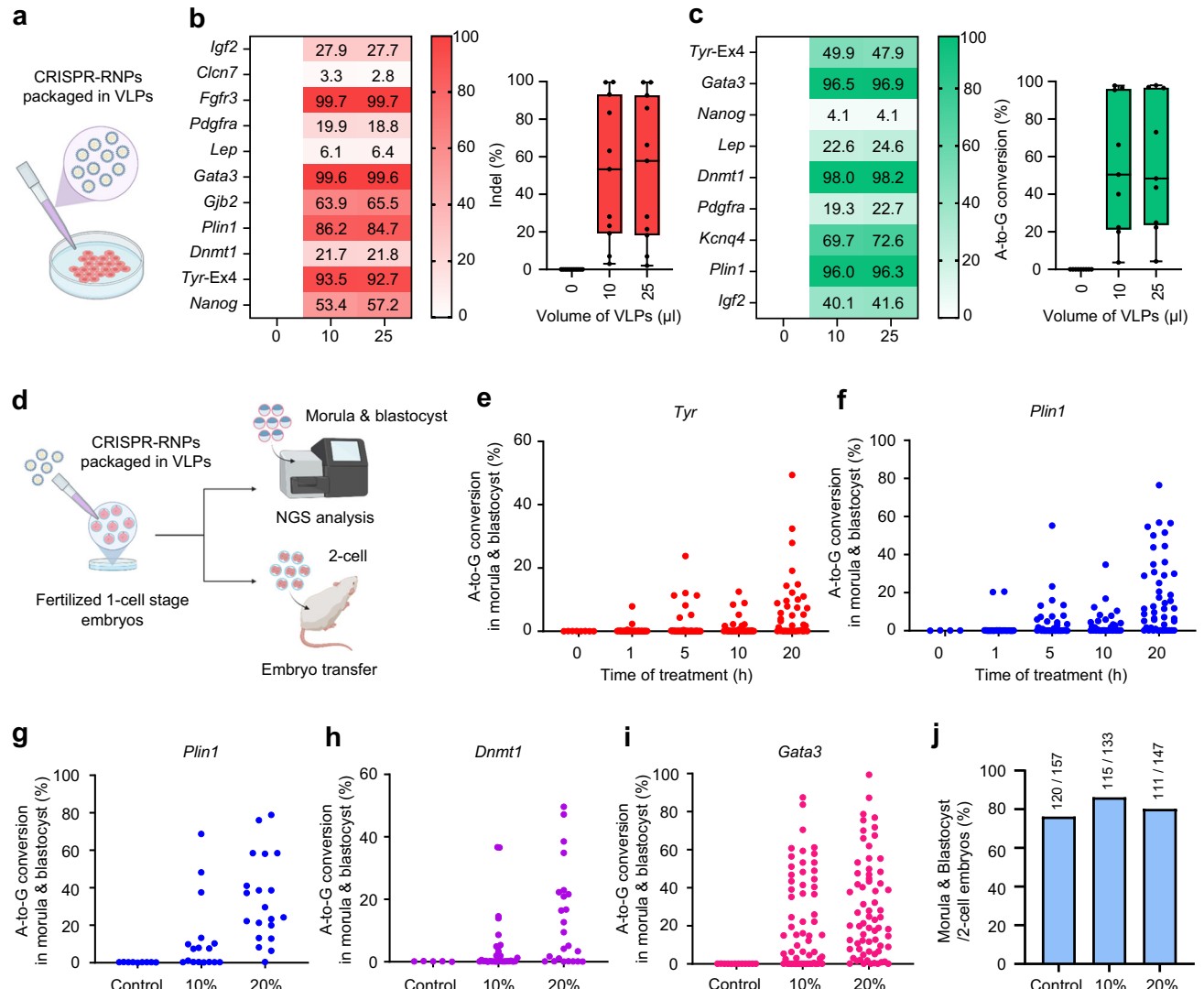

**Fig. 1 | Gene editing using the CRISPR-VIM method with CRISPR-RNP packaged in VLPs in mouse Neuro-2a cells and mouse zygotes. a** Scheme of gene editing using the CRISPR-VIM method in cells. Created in BioRender. Kim, K. (2025) https://BioRender.com/d01i280. **b** Generated indel frequencies at 11 mouse gene targets using SpCas9/sgRNA packaged in VLPs in Neuro-2a cells. **c** A-to-G conversion efficiencies for 9 targets obtained using ABE8e/sgRNA packaged in VLPs in Neuro-2a cells. Numbers in panels b and c represent mean values (*n* = 4). **d** Scheme of targeted mutagenesis of mouse zygotes via the CRISPR-VIM method. Created in BioRender. Kim, K. (2025) https://BioRender.com/e39e668. **e, f** Comparison of gene editing efficiencies based on treatment time (0, 1, 5, 10, and 20 h) in mouse

zygotes at *Tyr* (**e**) or *Plin1* (**f**) target site using ABE8e/sgRNA packaged in VLPs (*Tyr*: 10%, 1.97 × 10⁹ VLPs; 20%, 3.94 × 10⁹ VLPs; *Plin1*: 10%, 8.92 × 10⁸ VLPs; 20%, 1.78 × 10⁹ VLPs). **g–i** Gene editing efficiencies at *Plin1* (**g**), *Dnmt1* (**h**), and *Gata3* (**i**) targets were compared based on treatment rates (10% or 20% v/v VLPs) using ABE8e/sgRNA packaged in VLPs (*Plin1*: 10%, 8.92 × 10⁸ VLPs; 20%, 1.78 × 10⁹ VLPs; *Dnmt1*: 10%, 1.52 × 10⁹ VLPs; 20%, 3.04 × 10⁹ VLPs; *Gata3*: 10%, 1.53 × 10⁹ VLPs; 20%, 3.07 × 10⁹ VLPs). **j** Embryonic development rates at different treatment doses using ABE8e/sgRNA packaged in VLPs. The numbers above the bars indicate the counts of morula and blastocysts formed from all two-cell stage embryos. All data are shown as the mean ± SD. Source data are provided as a Source Data file.

concentrations (10% (+) and 20% (++)) across three stages: sperm pre-incubation, fertilization, and post-fertilization culture (Fig. 3a). The experimental group treated with 20% of VLP-packaged ABE8e/sgRNA after fertilization showed high editing efficiency of up to 76.7% for *Gata3* and up to 81.9% for *Plin1* (Fig. 3b, c and Supplementary Fig. 6g, h). Afterward, we checked the rate of morula and blastocyst development to determine VLP cytotoxicity during IVF. This evaluation was critical to determine the optimal stage for VLP application without compromising embryo viability. There was no significant difference in embryo development rates at both the morular and blastocyst stages between the control groups and the experimental group during post-fertilization culture. However, groups treated with VLPs during sperm pre-incubation or fertilization showed a significant decrease in embryo development rate compared to the other groups (Fig. 3d and Supplementary Fig. 9b).

Hence, applying the CRISPR-VIM method only during the post-fertilization culture stage is the most effective strategy for targeted mutagenesis in mice.

Moreover, we applied this method to create another mouse model. By targeting *Tyr*, gene editing efficiency was confirmed using 10% or 20% VLP-packaged ABE8e/sgRNA after IVF. In addition, transplanting 2-cell stage embryos into surrogate mothers provided mutant mice (Fig. 3e). We obtained *Tyr* mutant mice carrying the H420R genotype with an A-to-G substitution efficiency of 42.4% in the group treated with 20% of VLP-packaged ABE8e/sgRNA without off-target effects (Fig. 3f, g and Supplementary Figs. 3e, 4b). Furthermore, we identified germline transmission and Himalayan phenotype in the F2 generation of *Tyr*^H420R/H420R mutant mice[39] (Fig. 3h–k). This Himalayan phenotype confirmed the successful germline transmission and phenotypic appearance of the targeted mutation. Overall, these results

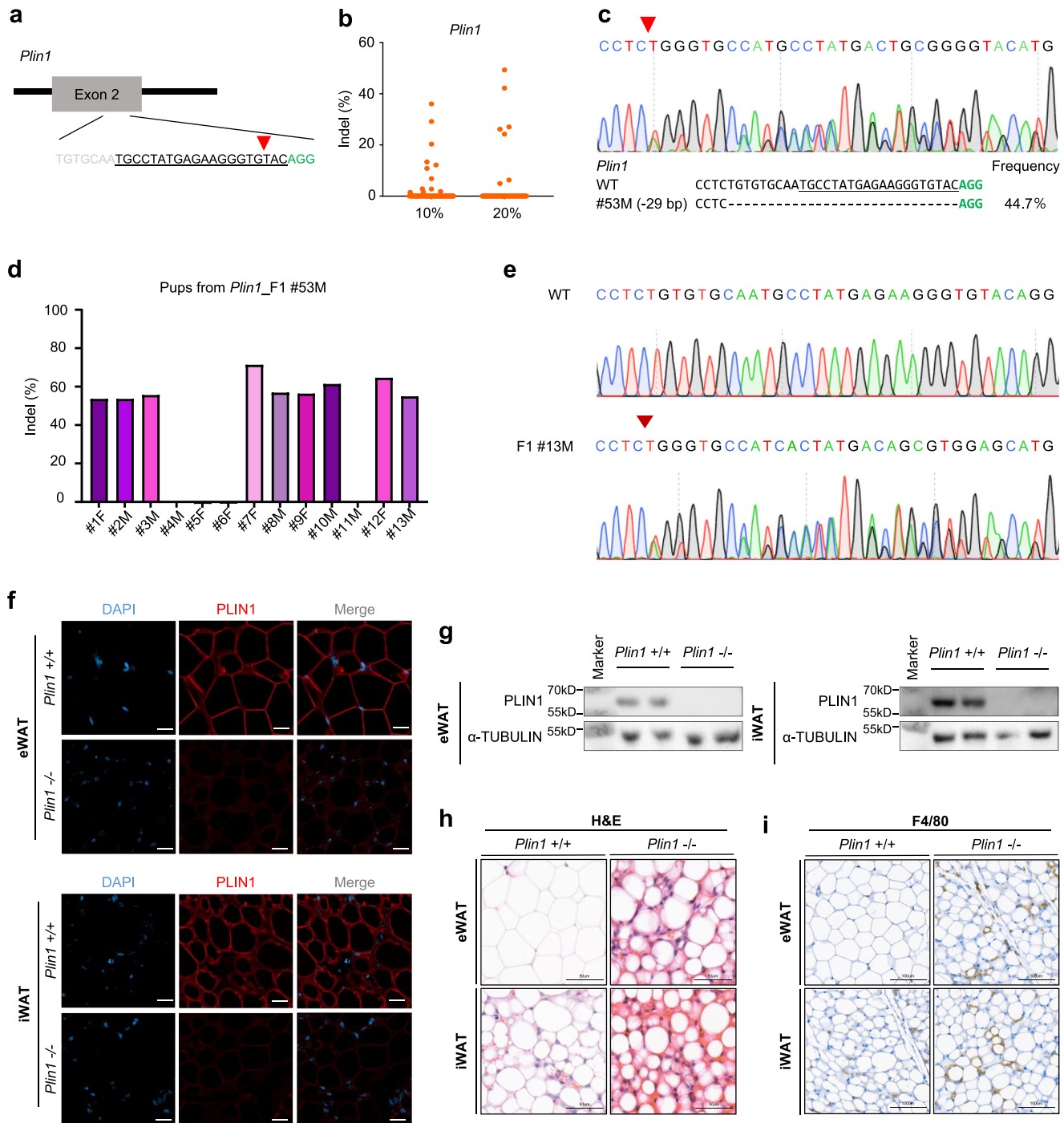

**Fig. 2 | Generation and phenotypic analysis of a *Perilipin1*-deficient mutant mouse model using Cas9/sgRNA packaged in VLPs. a** The target sequence at the *Perilipin1* (*Plin1*) locus. PAM sequences are shown in green, and sgRNA sequences are underlined in black, and the red arrowhead indicates the cleavage site. **b** Frequency of individual indels at the *Plin1* target site in mice treated with 10% or 20% Cas9/sgRNA packaged in VLPs, measured by targeted deep sequencing (10%, $1.50 \times 10^9$ VLPs; 20%, $3.00 \times 10^9$ VLPs). **c** Sanger sequencing chromatograms of a *Plin1*-knockout mouse with 29-base pair (bp) deletion. **d** Genotypes of the F1 generation obtained from F0 #53 M (*Plin1* mutant mouse with 29 bp deletion), determined by next-generation sequencing. **e** Sanger sequencing chromatogram of an F1 *Plin1*-knockout mouse containing 29 bp deletion. The red arrowhead represents deletion onset. **f** Immunofluorescence staining of PLIN1 (red) in epididymal white adipose tissue (eWAT) and inguinal WAT (iWAT) from *Plin1* (+/+) and *Plin1* (-/-) mice. Nuclei were counterstained with DAPI. Scale bars = 50 μm. **g** Immunoblot analysis of PLIN1 expression in eWAT and iWAT from *Plin1* (+/+) and *Plin1* (-/-) mice. **h** H&E staining in paraffin sections of eWAT and iWAT. Scale bars = 50 μm. **i** Immunohistochemistry staining of F4/80 in paraffin sections of eWAT and iWAT. Scale bars = 100 μm. *n* = 3 for *Plin1* (+/+) mice and *n* = 2 for *Plin1* (-/-) mice. Source data are provided as a Source Data file.

demonstrate that processing VLP-packaged CRISPR-RNPs during the post-fertilization culture stage is an efficient and reliable approach for generating targeted mutant mice with high gene editing efficiency and minimal off-target effects.

## Targeted mutagenesis through the CRISPR-VIM method based on codon-optimized CBE in cell lines and mouse zygotes

Then, we expanded the application of the CRISPR-VIM approach to CBE, a gene editing tool that has not been previously utilized in VLPs.

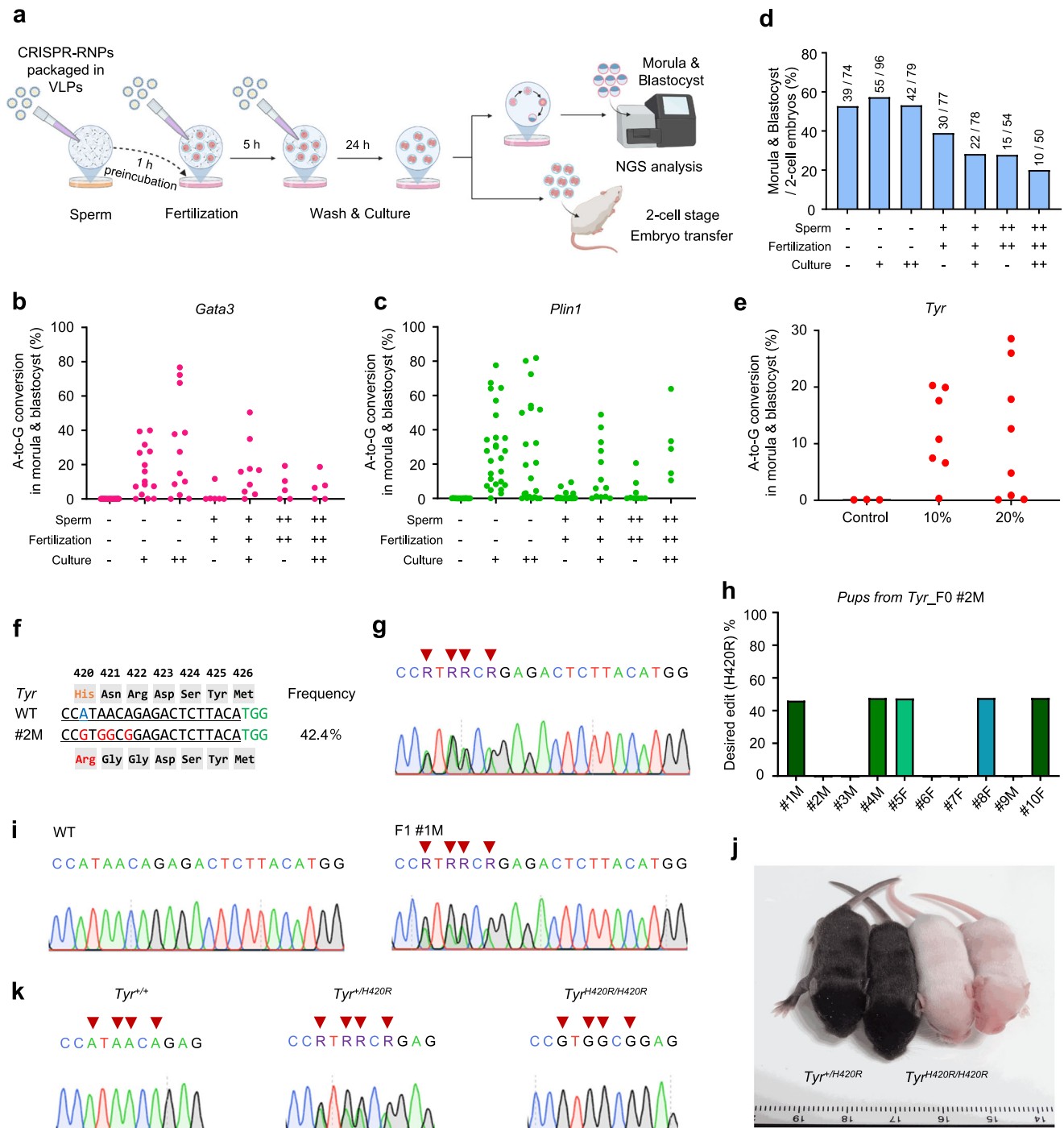

**Fig. 3 | Gene editing via the CRISPR-VIM method with CRISPR-RNP packaged in VLPs during IVF in mice. a** Scheme of gene editing via the CRISPR-VIM method during IVF in mice. Created in BioRender. Kim, K. (2025) https://BioRender.com/b91u430. **b**, **c** Comparison of gene editing efficiencies at *Gata3* (**b**) and *Plin1* (**c**) target sites in groups treated with ABE8e/sgRNA packaged in VLPs during IVF (*Gata3*: 10%, $1.53 \times 10^9$ VLPs; 20%, $3.07 \times 10^9$ VLPs; *Plin1*: 10%, $8.92 \times 10^8$ VLPs; 20%, $1.78 \times 10^9$ VLPs). **d** Embryo development rates in experimental groups treated with ABE8e/sgRNA packaged in VLPs during IVF. Numbers above the bars in the graph indicate the number of morula and blastocysts formed from two-cell stage embryos. +, 10% of CRISPR-RNP packaged in VLPs of total medium volume; ++, 20% of CRISPR-RNP packaged in VLPs of total medium volume. **e** A-to-G conversion efficiency in morula and blastocysts cultured in vitro after treatment with ABE8e/sgRNA packaged in VLPs targeting *Tyr* post-fertilization stage. 10%, $1.14 \times 10^9$ VLPs;

20%, $2.27 \times 10^9$ VLPs. **f** Mutation pattern of representative *Tyr*-mutant mouse #2 M with the H420R mutation obtained using ABE8e/sgRNA packaged in VLPs via IVF. **g** Sanger sequencing chromatogram of *Tyr* mutant mouse #2 M. Red arrowheads indicate A-to-G conversion site. **h** Genotyping of the *Tyr* (H420R) F1 generation by next-generation sequencing. **i** Sanger sequencing chromatograms of a WT mouse and an F1 #1 M (*Tyr*^+/H420R^) mutant mouse. The red arrowhead represents A-to-G conversion. **j** Coat color change of *Tyr*^H420R/H420R^ mutant F2 mice showing the Himalayan phenotype. **k** Sanger sequencing of *Tyr* wild-type (*Tyr*^+/+^), heterozygous (*Tyr*^+/H420R^), and homozygous (*Tyr*^H420R/H420R^) mutant F2 mice. +, 10% of CRISPR-RNP packaged in VLPs of the total culture medium volume; ++, 20% of CRISPR-RNP packaged in VLPs of the total culture medium volume. All data are shown as the mean ± SD. Source data are provided as a Source Data file.

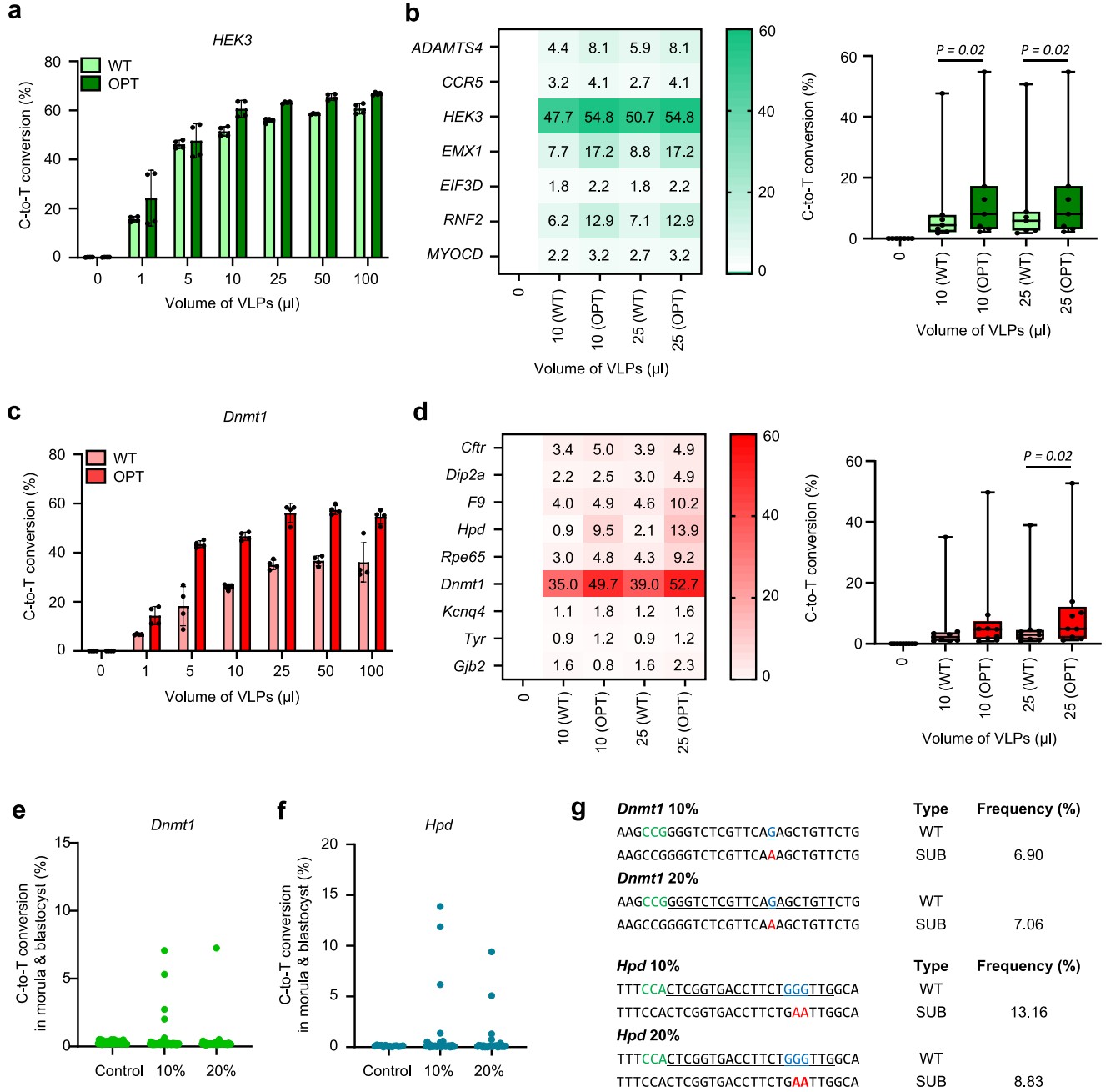

**Fig. 4 | Base editing using CBE packaged in VLPs in cell lines and mouse zygotes.** **a** Optimization of processing conditions for AncBE4max/sgRNA (CBE) packaged in existing VLPs (WT) and optimized VLPs (OPT), targeting *HEK3* in HEK293T cells (*n* = 4). **b** Comparison of C-to-T conversion efficiencies using CBE packaged in WT or OPT VLPs targeting human genes (*ADAMTS4, CCR5, HEK3, EMX1, EIF3D, RNF2*, and *MYOCD*) in HEK293T cells (*n* = 4, *p* = 0.02, the statistical test used was two-tailed). Data is presented as a box plot with median value. The borders of box represent the first and the third quarter percentiles and the whiskers indicate the highest and lowest value. **c** C-to-T conversion efficiency at *Dnmt1* target using CBE packaged in WT or OPT VLPs at varying volumes in mouse Neuro-2a cells (*n* = 4). **d** Comparison of editing efficiencies using CBE packaged in WT or OPT VLPs across various mouse targets (*Cftr, Dip2a, F9, Hpd, Rpe65, Dnmt1, Kcnq4, Tyr*, and *Gjb2*) (*n* = 4, *p* = 0.02 the

statistical test used was two-tailed). Data is presented as a box plot with median value. The borders of box represent the first and third quater percentiles and the whiskers indicate the highest and lowest value. **e**, **f** C-to-T conversion efficiencies mediated by CRISPR-VIM in mouse embryos targeting *Dnmt1* and *Hpd* using optimized VLP-packaged CBE (*Dnmt1*: 10%, $2.31 \times 10^9$ VLPs; 20%, $4.62 \times 10^9$ VLPs; *Hpd*: 10%, $2.15 \times 10^8$ VLPs; 20%, $4.30 \times 10^9$ VLPs). **g** Mutation patterns and editing efficiency (wild-type: WT, substitution: SUB) in mouse embryos induced by the CRISPR-VIM method with CBE packaged in VLPs. Statistical significance was determined using a paired *t* test. All data are shown as the mean ± SD. The bar plot values correspond to the mean value of all biological replicates, and the error bars illustrate the SD. **a**–**d** Source data are provided as a Source Data file.

We used AncBE4max[40], an optimized version of CBE, packaged it into VLPs, and applied it to human HEK293T and mouse Neuro-2a cells. The C-to-T conversion efficiency of up to 50.7% for the human target *HEK3* and up to 39.0% for the mouse target *Dnmt1* was observed by treating cell lines with 25 µl CBE/sgRNA packaged in VLPs (Fig. 4a–d). However, CBE/sgRNA packaged in VLPs did not have higher overall C-to-T

conversion efficiency in both human and mouse targets compared to SpCas9/sgRNA packaged in VLPs and ABE8e/sgRNA packaged in VLPs. To overcome this issue, we aimed to enhance the efficiency of C-to-T conversion by improving the VLP packaging of MLV gag polyprotein in MMLVgag-3xNES-AncBE4max through human codon optimization. By packaging CBE with human codon-optimized VLPs (CBE (OPT)), we

increased the C-to-T conversion efficiency by up to 2.3-fold for human targets and up to 6.5-fold for mouse targets (Fig. 4a–d and Supplementary Fig. 5a, b). CBE/sgRNA packaged in VLPs contained a FLAG tag at the Cas9 N-terminus, allowing us to quantify Cas9 levels via ELISA. This analysis revealed that CBE (OPT) contained approximately twice the amount of Cas9 compared to CBE (WT), highlighting the improved loading efficiency of the optimized VLP system (Supplementary Fig. 5c). We also applied the optimized CBE-based VLPs to mouse zygotes, showing successful C-to-T conversion efficiencies of up to 7.3% for *Dnmt1* target and up to 13.9% for *Hpd* target (Fig. 4e–g and Supplementary Figs. 6j, k, 9a). These results suggest that the codon-optimized VLP-packaged CBE system can be used as a practical strategy for generating mouse models.

### Knock-in via SpCas9/sgRNA-based CRISPR-VIM method using adeno-associated virus donor in mESCs and mouse zygotes

In the mouse model generation strategy using CRISPR-Cas9/sgRNA-based HDR-mediated knock-in, the persistence of donor DNA is directly related to the knock-in efficiency. Here, we employed an innovative knock-in strategy based on CRISPR-VIM, packaging donor DNA into adeno-associated virus (AAV)[33,41] and co-culturing VLP-packaged CRISPR-RNPs with cells and zygotes.

To optimize the AAV donor system, we compared single-stranded AAV (ssAAV) and self-complementary AAV (scAAV) with AAV/6 or AAV/DJ serotypes. Given its self-complementary structure and higher systematic stability, scAAV demonstrated higher efficiency in delivering EGFP in both mouse cells and zygotes, leading to its selection as the AAV donor in subsequent experiments. In addition, we used scAAV/6 serotype as the AAV donor in CRISPR-VIM-based knock-in (Supplementary Fig. 7). To determine the optimal treatment time for AAV donors, we performed a comparative analysis between two strategies: strategy 1 involved treating both VLPs and AAV donors simultaneously, while strategy 2 involved treating AAV donors 6 hours before introducing CRISPR-RNP packaged in VLPs (Supplementary Fig. 8a). We also systematically tested different AAV concentrations to identify the most effective dose. As a knock-in target, we induced knock-in of *Eco*RI recognition sequence in exon 1 of mouse *Tyr* gene (Supplementary Fig. 8b). Experimental validation in Neuro-2a cells and mESCs showed notable knock-in efficiencies of up to 13% and 35%, respectively[42] (Supplementary Fig. 8c–e). Comparative analysis of VLP treatments identified no substantial differences between simultaneous co-treatment and pre-treatment with AAV donors 6 h before the VLP treatment. Both strategies consistently achieved optimal efficiency with a standardized AAV concentration of $1 \times 10^5$ MOI (Supplementary Fig. 8c, d). To validate knock-in efficiency, the target sequence was first analyzed through Sanger sequencing, followed by PCR amplification and *Eco*RI enzyme digestion. Both methods confirmed the precise integration of the *Eco*RI recognition sequence in the mouse *Tyr* gene (Supplementary Fig 8e, f).

Based on these results, we applied the same method to mouse zygotes to create genetically humanized mouse model (Fig. 5a). We selected mouse *Kcnq4* gene and designed human *KCNQ4* knock-in sequences of 728 bp (Fig. 5b). To generate a genetically humanized *KCNQ4* exon 5 knock-in mouse model, we used a more highly concentrated VLP system to target two specific sites for exon exchange. This approach resulted in successful genetic humanization in 2 out of 8 blastocysts and 1 out of 12 F0 offspring, with stable germline transmission confirmed in subsequent generations (Fig. 5c, d, f, and Supplementary Fig. 9c). These results indicate that the CRISPR-VIM method is a versatile tool capable of inducing knock-in for both small insertions, such as enzyme site, and large insertions over 700 bp. Thus, the CRISPR-VIM HDR approach offers a highly efficient and simplified method for generating knock-in models, with broad applicability for creating both small and large genomic modifications.

### Multi-target gene editing of cell lines and mouse zygotes using the CRISPR-VIM method

A previous study confirmed the feasibility of multi-target gene editing in a cell line using CRISPR RNP-based VLPs[27,30]. We performed gene editing by applying different treatment approaches (combi, double, and triple) using VLP-packaged SpCas9/sgRNA or ABE8e/sgRNA into mouse Neuro-2a cells and zygotes to evaluate the potential for multi-gene targeting (Fig. 6a). The combi approach packages two sgRNAs into a single VLP, simultaneously targeting of two loci. However, this approach does not allow the combined use of Cas9 and ABE8e, as neither tool can selectively distinguish between individual sgRNAs, which may result in unintended mutations. In double or triple approach, each sgRNA is packaged into a separate VLP, allowing simultaneous delivery of two or three VLPs for multi-targeting (Fig. 6a). We delivered combi, double or triple approach using VLP-packaged CRISPR-SpCas9/sgRNA or ABE8e/sgRNA in mouse Neuro-2a cells or zygotes and identified indel or A-to-G conversion mutation efficiencies (Fig. 6b–j, and Supplementary Fig. 9d). Despite lower editing efficiency compared to the combi approach, these strategies effectively support multi-target editing, particularly when using different gene editing tools. Expanding beyond the triple approach, the quadruple approach was introduced, enabling the simultaneous induction of diverse editing formats in cells. This approach combined VLP-packaged SpCas9/sgRNA, AncBE4max/sgRNA, and ABE8e/sgRNA achieving notable efficiencies: up to 88.7% in *Fgfr3* and 24.5% in *Kcnq4*, 21.8% in *Dnmt1*, and 42.3% in *Gata3* (Fig. 6f). In conclusion, the CRISPR-VIM method demonstrates robust potential for multi-target editing in both cell lines and mouse embryos.

Next, we created mouse models by applying multi-target gene editing using CRISPR-RNP packaged in VLPs, with both combi and triple approaches. Among the 82 mice produced through the combi approach, 6 carried mutations in both targets, representing the most notable outcome (Fig. 6k).

In addition, 24 mice had mutations in only one of the targets. The triple approach resulted in 59 mice, none of which had mutations in all three targets. However, 11 had mutations in one target, and 1 carried mutations in two targets, highlighting the reduced efficiency of the triple approach for multi-target editing in mice. These results indicate that while the combi and triple approaches enable multi-target editing, efficiency decreases with an increasing number of targets. For mouse embryos, limiting the number of targets to two proved more effective. Furthermore, we used mice born from the combi approach to evaluate the number of generations required to produce a homozygote genotype for both mutations. In F0 mice with two mutations, we observed that if each target had a mutation rate exceeding 10%, the mutations were predominantly transmitted to the F1 generation (Fig. 6l). Through crossbreeding heterozygous F1 mice carrying both mutations (male: F1 # 5 M with *Gata3* at 47.21 % and *Kcnq4* at 48.33 % was crossed with female: F1 # 1 F with *Gata3* at 45.72 % and *Kcnq4* at 47.40% or F1 # 9 F with *Gata3* at 45.71 % and *Kcnq4* at 47.81 %), we generated 17 F2 mice, one of which was homozygous for both targets (Fig. 6m, n, o). These findings confirm that the genetically modified mouse model generated through multi-target editing using the VLP delivery system was stably transmitted across generations, with homozygous mutants successfully obtained in the F2 generation.

## Discussion

We introduced the CRISPR-VIM method, an innovative approach for embryonic genome editing and animal model generation. The CRISPR-VIM method overcomes the limitations of existing embryo editing technologies, including the dependency on highly skilled researchers and specialized microinjection or electroporation equipment. Moreover, this versatile mouse model production method can be applied not only to the use of mouse zygotes but also to IVF. The CRISPR-VIM method utilizes previously established Cas9- and ABE-based VLPs in

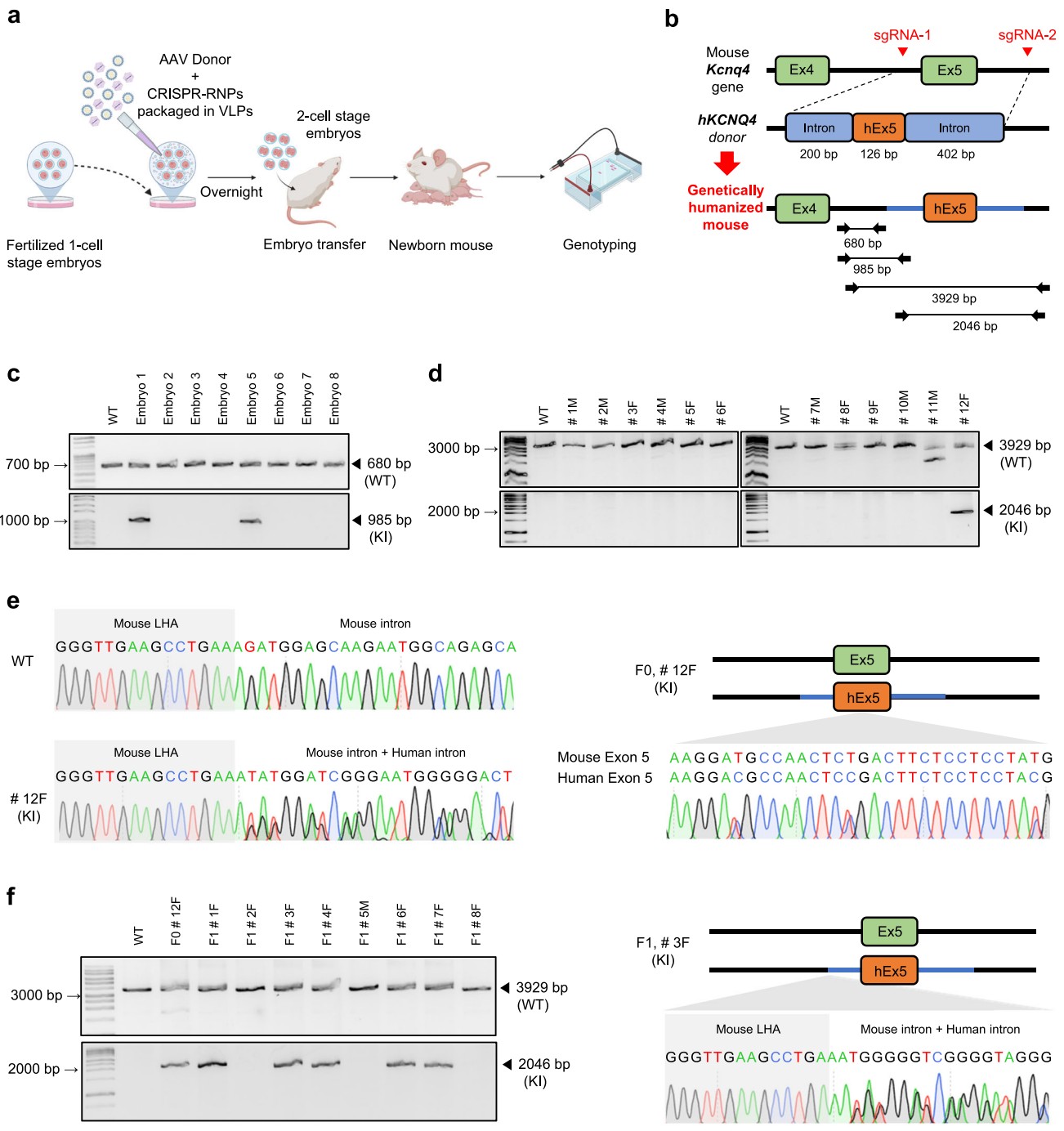

**Fig. 5 | HDR-mediated knock-in via the CRISPR-VIM method with SpCas9/ sgRNA packaged in VLPs. a** Schematic of the CRISPR-VIM-based HDR strategy with AAV-based donors in mouse embryos. Created in BioRender. Kim, K. (2025) https:// BioRender.com/a99d722. **b** Strategy for generating a knock-in mouse model where exon 5 and flanking intronic sequences (a total of 728 bp) of the mouse *Kcnq4* gene are replaced with human *KCNQ4* (h*KCNQ4*) pathogenic mutation and sequences. **c** PCR-based genotyping of *Kcnq4* knock-in mouse embryos, confirming the introduction of human sequences via the CRISPR-VIM-based HDR approach.

**d** Genotyping analysis of F0 mice demonstrating the successful knock-in of human sequences at the *Kcnq4* locus. **e** Sanger sequencing of F0 mice validating the precise knock-in of human sequences into the *Kcnq4* locus. **f** Germline transmission analysis of humanized *KCNQ4* mouse genotypes, shown via gel electrophoresis (left) and Sanger sequencing (right). WT indicates wild-type, and KI indicates knock-in. All experiments were repeated three times independently. Source data are provided as a Source Data file.

cells and mouse zygotes. Using VLP-packaged SpCas9/sgRNA or ABE8e/sgRNA RNP, *Plin1*-knockout mice or *Tyr* H420R mutant mice were generated without off-target effects, and their respective phenotypes were observed and analyzed. Furthermore, the CRISPR-VIM method not only allows for the simultaneous editing of two or more gene targets but also excels in its suitability for the generation of more sophisticated mouse models with complex genetic modifications. In

addition, it enables CBE-mediated C-to-T substitution and HDR-mediated knock-in gene editing.

Specifically, the knock-in strategy via CRISPR-VIM enables stable delivery of donor DNA using the scAAV system and can easily generate knock-in models. Nonetheless, ongoing efforts are required to optimize HDR-mediated knock-in and CBE editing, particularly to address the relatively low developmental rate observed when AAV donors are

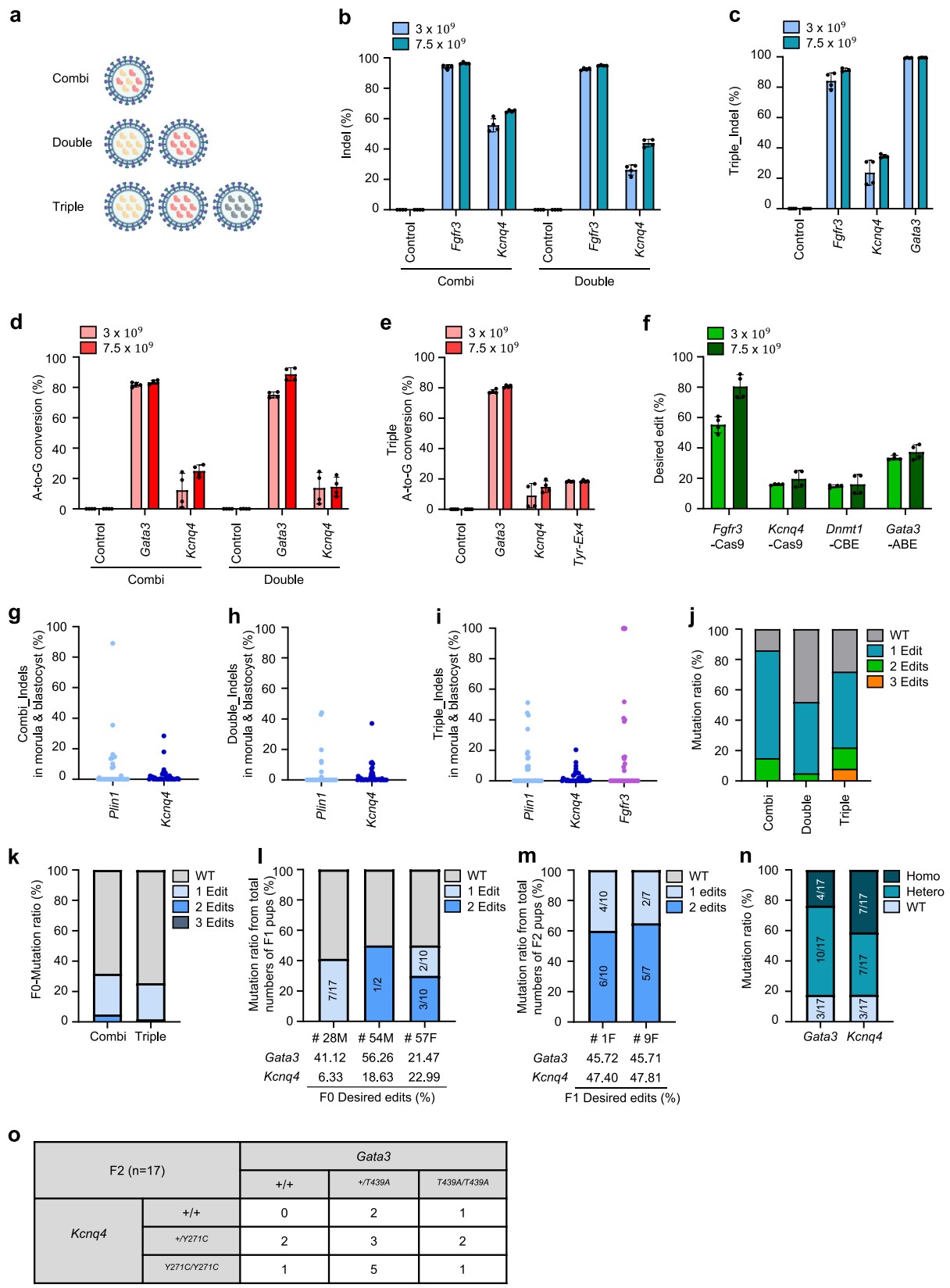

| F2 (n=17) | | Gata3 | | |
|---|---|---|---|---|
| | | +/+ | +/T439A | T439A/T439A |
| Kcnq4 | +/+ | 0 | 2 | 1 |
| | +/Y271C | 2 | 3 | 2 |
| | Y271C/Y271C | 1 | 5 | 1 |

combined with VLP treatment. Engineering VLPs to improve their transduction efficiency in embryos is a promising approach to enhance the efficiency of GEMM generation using the CRISPR-VIM method.

Furthermore, engineering VLPs to concurrently package donor DNA enables a more efficient generation of knock-in mice. While VLPs are known to elicit a low immune response, further capsid engineering is crucial to completely eliminate potential immune responses. Moreover, advancing VLP systems to enable precise targeting of specific tissue or cell types will significantly broaden their versatility and enhance their potential in therapeutic and research applications.

In this study, we developed an innovative CRISPR-VIM method that represents a significant advancement in animal model generation

**Fig. 6 | Multi-target editing using the CRISPR-VIM method with CRISPR-RNPs packaged in VLPs. a** Schematic of multi-target editing via CRISPR-VIM. The combi approach involved packaging two sgRNAs into a single VLP, while the double and triple approaches used separate VLPs for each sgRNA, combining two or three different VLPs to achieve simultaneous gene editing. Created in BioRender. Kim, K. (2025) https://BioRender.com/c06n497. **b** Editing efficiencies of combi and double approaches targeting *Fgfr3* and *Kcnq4* in mouse Neuro-2a cells with SpCas9/sgRNA packaged in VLPs (*n* = 4). **c** Triple gene editing efficiencies targeting *Fgfr3*, *Kcnq4*, and *Gata3* in mouse Neuro-2a cells with SpCas9/sgRNA packaged in VLPs (*n* = 4). **d** Combi gene editing efficiencies using VLP-packaged ABE8e/sgRNA targeting *Gata3* and *Kcnq4* in mouse Neuro-2a cells (*n* = 4). **e** Triple gene editing for A-to-G base conversion at *Gata3*, *Kcnq4*, and *Tyr*-Ex4 loci using ABE8e/sgRNA in VLPs (*n* = 4). **f** Quadruple gene editing efficiencies, showing both desired base substitutions and indel formation at four loci (*n* = 4). **g–i** Editing efficiencies in mouse embryo using (**g**) combi gene editing (*Plin1*+*Kcnq4*; 20%, 2.57 × 10⁹ VLPs), (**h**) double gene editing (*Plin1*, *Kcnq4*; 1.50 × 10⁹ VLPs), and (**i**) triple gene editing (*Plin1*, *Kcnq4*, *Gata3*; 1.00 × 10⁹ VLPs) with SpCas9/sgRNA packaged in VLPs. **j** Gene editing ratio in mouse embryos from combi, double, and triple gene editing approaches. **k** Genotype distributions of F0 mice produced by combi and triple gene editing. **l, m** Genotype distributions of the (**l**) F1 and (**m**) F2 generations from F0 mice generated via the combi gene editing approach. F1 mice were bred by crossing mosaic and wild-type mice, and F2 mice were generated by crossing heterozygous F1 mice. **n** Genotype distributions of F2 mice generated through the combi gene editing approach. Numbers within the bars indicate the count of each genotype among all offspring. **o** Genotype distributions of F2 mice at *Gata3* and *Kcnq4* loci. Bar plots show mean values ± SD. **b–f** All data are presented as mean ± SD. Source data are provided as a Source Data file.

while addressing various challenges associated with traditional techniques, such as microinjection and electroporation. This method has the potential to be applied not only to mice but also to medium- and large-sized animals. However, further research is required to validate its applicability to medium- and large-sized animals. Therefore, the CRISPR-VIM method presents various possibilities as a potential method for embryonic gene editing and the creation of animal models.

## Methods

### Ethics
Animal experiments were performed in accordance with protocols approved by the Institutional Animal Care and Use Committee (IACUC) of Seoul National University (Approval no. SNU-220930-2-1, SNU-230813-1-1) and Korea University (Approval no. KOREA-2022-0105-C2, KOREA-2022-0105).

### Vector cloning
For sgRNA-containing plasmids, target spacer sequences were synthesized and ligated with pRG2-GG (Addgene, 104174) using T4 ligase (New England Biolabs, M0202L). pBS-CMV-gagpol (35614), pCMV-MMLVgag-3xNES-Cas9 (181752), and pCMV-MMLVgag-3xNES-ABE8e (181751) plasmids were obtained from Addgene. pCMV-MMLVgag-3xNES-AncBE4max and -AncBE4max (OPT) were cloned via the Gibson assembly method using HiFi DNA Assembly Master Mix (NEB, E2621). The plasmids were transformed into competent cells (*Escherichia coli* DH5α; Enzynomics, CP010). The amplified plasmids were extracted using the NucleoBond Xtra Midi Kit (Macherey-Nagel, 740410.1) according to the manufacturer's protocol.

### Production and purification of VLP-packaged CRISPR-RNP
VLPs were produced by Gesicle producer 293 T cells (Takara, 632617). Briefly, 5 × 10⁶ cells were seeded in 75 T flasks (Corning, 34322023) and cultured in Dulbecco's Modified Eagle Medium (DMEM) (Welgene, LM001-05) containing 10% fetal bovine serum (FBS; Welgene, S101-01). After 24 h of cell seeding, the cells were transfected using the jetPRIME transfection reagent (Polyplus, 101000001). The transfection reagent was mixed with VSV-G (400 ng), pBS-CMV-gagpol (3375 ng), MMLVgag-3xNES-ABE8e, -Cas9, -AncBE4max, or -AncBE4max (OPT) (1125 ng), and sgRNA (4400 ng)³⁰. The transfection reagent and plasmids were co-transfected into cells according to the manufacturer's protocol. After 24 h of transfection, the cell culture medium was replaced with 20 ml of fresh medium. After 48 h of transfection, the supernatant was harvested into a conical tube and centrifuged at 500 × g for 5 min using an Allegra X-15R centrifuge (Beckman coulter). Afterward, the harvested supernatants were filtered using a 0.45 μm polyvinylidene fluoride filter syringe (Daihan Scientific, DH.Fil 3066) or a vacuum bottle-top filtration system (Millipore, S2HVU02RE). Subsequently, the filtered supernatants were mixed with 5X PEG-it Virus Precipitation Solution (SBI, LV810A-1) according to the manufacturer's protocol and stored overnight at 4 °C. The concentrated solution was centrifuged at 1500 × g at 4 °C for 30 min. For knock-in experiments in mouse zygotes, the filtered supernatants were concentrated to a higher concentration by centrifugation at 32,240 × g in 4 °C for at least 1 h, using 20% (w/v) sucrose (Sigma, S0389-500G). The precipitates containing VLP-packaged CRISPR-RNP were resuspended in chilled phosphate-buffered saline (PBS) (Welgene, ML008-01) or Opti-MEM (Gibco, 31985-062). Then, the produced VLPs were stored at − 80 °C.

### Titration of VLP-packaged CRISPR-RNP
VLP titers were quantified using the MuLV Core Antigen ELIZA kit (Cell Biolabs; VPK-156) following the manufacturer's protocol. Based on a previous study, one VLP was calculated to contain 1800 molecules of p30, and 20% of p30 is associated with VLP³⁰,⁴³. Protein contents of Cas9 containing VLPs were measured using FLAG-Tag (DYKDDDDK-Tag protein) ELIZA kit (Finetest, EU2607). VLP values for all targets can be found in Supplementary Fig. 10.

### Cell culture and VLP treatment
ARPE-19 (ATCC, CRL-2302) and HEK293T (ATCC, CRL-3216) cells were maintained in DMEM (Welgene, LM001-05) containing 10% FBS (Welgene, S101-01). Neuro-2a cells (ATCC, CCL-131) were cultured in MEM (Gibco, 10370021) containing 10% FBS, 1% sodium pyruvate, and 1% L-glutamine. mESCs were cultured in serum-free ES medium (SFES) containing PD03259010 (Stemgent, 04-0006), CHIR99021 (Stemgent, 04-0004), glutamine (Gibco, 25030-081), monothioglycerol (Sigma, M6145-25ML), and LIF (Escgro, ESG1106). The SFES was composed of NEUROBASAL (Gibco, 21103-049), DMEM/F12 (Gibco, 11320-033), N2-Supplement (Gibco, 17502-048), B27 + RA (Gibco, 17504-044), 7.5% bovine serum albumin (BSA; Gibco, 15260-037), and penicillin-streptomycin (Welgene, LS202-02). All cell lines were maintained at 37 °C with 5% CO₂. For treating cells with VLP-packaged CRISPR-RNP, 3 × 10⁴ cells were seeded per well in 48-well plates (SARSTEDT, 83.3923) containing 500 μl of complete media. mESCs were seeded with 1 × 10⁴ cells per well in gelatin-coated 48-well plates. After 24 h of cell seeding, the required volume of VLP-packaged CRISPR-RNP was added to the wells containing the cells. At 72 hours of post-treatment, the cells were harvested, and genomic DNA was extracted.

### Animals
C57BL/6 N male and female mice were purchased from DBL CO., Ltd. (South Korea) and ORIENT BIO Inc. (South Korea). Female ICR mice from the same suppliers were used as surrogate mothers. Unless otherwise specified, all experiments were conducted using C57BL/6 N mice, except in cases involving surrogate mothers. Female mice aged 5–8 weeks and male mice aged 10–26-week were used for experiments. For in vitro fertilization (IVF), 5-6-week-old C57BL/6 N female mice and 11–16-week-old C57BL/6 N male mice were used. All mice were maintained in a specific pathogen-free (SPF) facility under a 12 h light/dark cycle, with a controlled temperature of 20–26 °C and humidity levels of 40–60%.

## Zygote preparation and VLP treatment

Preparation of mouse zygotes for the CRISPR-VIM method involved the following steps. First, 5–8-week-old female C57BL/6 N mice were superovulated via an intraperitoneal injection of HyperOva (CARD, KYD-010-EX x5) and human chorionic gonadotropin (hCG; Sigma-Aldrich, CG10-1vl) at a 48 h interval[44]. Then, these mice were mated with 10–26-week-old C57BL/6 N male mice. Zygotes were collected from the oviducts and placed in the M2 medium (Sigma-Aldrich, M7167). Cumulus cells were removed from zygotes by exposing them to 0.1% hyaluronidase (Sigma-Aldrich, H3884) in PBS[45]. Zygotes were incubated in a total medium volume of 50 μl containing potassium simple optimized medium (KSOM; MR-121-D, Millipore) with VLP-packaged CRISPR-RNPs at 37 °C in a humidified atmosphere containing 5% $CO_2$. Following incubation, 2-cell stage embryos were transplanted into the oviducts of 0.5-day-post-coitus pseudo-pregnant surrogated mothers to obtain offspring[39]. For in vitro analysis, untreated and VLP-packaged CRISPR-RNP–treated zygotes were cultured in KSOM for 4 days to obtain morula and blastocyst.

## Application of CRISPR-VIM in IVF

We extracted spermatozoa clots from the cauda epididymis of 11–16-week-old male C57BL/6 N mice and cultured them for 1 h at 37 °C in a pre-incubation medium (CARD, KYD-002-05-EX) containing VLP-packaged CRISPR-RNP. Subsequently, 5-6-week-old female C57BL/6 N mice were superovulated via an intraperitoneal injection of HyperOva and hCG at a 48 h interval. Oocytes were collected from the oviducts and placed in the HTF medium (Cosmo Bio, CSR-R-B070) treated with VLP-packaged CRISPR-RNP. Subsequently, the collected oocytes were inseminated with sperm obtained from the pre-incubation medium at 37 °C in a humidified atmosphere containing 5% $CO_2$ for 5 h. After fertilization, the oocytes were washed twice in the HTF medium, followed by a single wash in the HTF medium diluted with VLP-packaged CRISPR-RNP at concentrations of 0%, 10%, or 20% in each group. The oocytes were incubated overnight in the HTF medium with the same concentration in each group at 37 °C in a humidified atmosphere containing 5% $CO_2$. After overnight culture, 2-cell stage embryos were transplanted into surrogate mothers in the same manner as performed for zygotes. We cultured 2-cell stage embryos in KSOM for an additional 3 days to obtain morula and blastocyst for in vitro analysis.

## scAAV production

For the expression of the scAAV (scAAV overexpressing EGFP), the pscAAV-CMV-GFP vector (#32396, Addgene) construct was purified, with capsids from AAV serotype 6 and DJ, through the generation of scAAV utilizing a helper-free HEK293 cell system. The purification process involved iodixanol gradient ultracentrifugation, conducted at the KIST Virus Facility. The production titer, determined by qPCR, were $2.02 \times 10^{13}$ GC/ml (Genome Copy/ml) for pscAAV6-CMV-GFP, $1.24 \times 10^{13}$ GC/ml (Genome Copy/ml) for pscAAVDJ-CMV-GFP, $2.59 \times 10^{13}$ GC/ml (Genome Copy/ml) for pscAAVDJ-Tyr-EcoRI, and $1.10 \times 10^{12}$ GC/ml for pscAAV6-Kcnq4, respectively.

## Genotyping via targeted deep sequencing, Sanger sequencing, and enzyme digestion

Genomic DNA was extracted from the cells, mouse embryos, and neonatal mouse tissues via treatment with a lysis buffer (25 mM NaOH and 0.2 mM EDTA in distilled water) or using the DNeasy Blood & Tissue Kit (Qiagen, 69506) according to the manufacturer's protocol. Genomic DNA was used to amplify the target gene sites using SuperFi II PCR Master Mix (Invitrogen, 12368050), KAPA polymerase (Roche, 7958897001), or SUN PCR Blend (SUN GENETICS, SG-PT02). The PCR samples were subjected to paired-end sequencing using Miseq and MiniSeq instruments (Illumina). Obtained data were analyzed using the EUN program (https://daeunyoon.com/)[44,46]. For Sanger sequencing, PCR amplicons were purified using Expin™ PCR SV (GeneAll, 103-102)

according to the manufacturer's protocol and analyzed by Bionics. The Sanger sequencing results were analyzed using the ICE program (https://ice.synthego.com/). Enzyme digestion was conducted using EcoRI-HF (NEW ENGLAND BioLabs, R3101S) restriction enzyme with purified PCR products according to the manufacturer's protocol. The primer sets used are listed in Supplementary Tables 1 and 2.

## Off-target analysis

Cas-OFFinder available in the CRISPR RGEN Tools (http://www.rgenome.net/) was used to identify potential off-target candidates in the Mus musculus (GRCm38/mm10) genome while allowing mismatches of up to three base pairs[47]. The gene editing efficiencies for on- or off-target were analyzed using next-generation sequencing. The sequences of the off-target candidates and primer sets used are listed in Supplementary Tables 3 and 4.

## Western blotting

eWAT and iWAT depots of wild-type and Plin1-knockout mice were lysed and extracted using RIPA buffer (BIOSOLUTION, BR002) containing phosphatase inhibitor (GenDEPOT, P3200-005) and protease inhibitor cocktail (GenDEPOT, P3100-005). Total protein lysates were boiled with 5X Laemmli sample buffer (ELPIS-BIOTECH, EBA-1052), separated by sodium dodecyl sulfate-polyacrylamide gel electrophoresis, and transferred to a PVDF membrane (Invitrogen, IB34001). The PVDF membrane blots were blocked in 5% BSA (GenDEPOT, A0100-010) in Tris-buffered saline with Tween20 (TBS-T, Biosesang, TR2007-000-74) and incubated at 4 °C overnight with diluted primary antibodies (Perilipin 1, Invitrogen, PA1-1051, 1:1,000; α-Tubulin, Abbkine, A01080, 1:1,000). Mouse anti-rabbit IgG-HRP (Santa Cruz Biotechnology, sc-2357, 1:5,000) and m-IgGκ BP-HRP (Santa Cruz Biotechnology, sc-516102, 1:5,000) were used as secondary antibodies. Protein expression was detected using Clarity Western ECL Substrate (Bio-Rad, 1705061) and visualized using Chemi-Doc XRS+ System (Bio-Rad)[48].

## Immunohistochemistry and immunofluorescence

eWAT and iWAT from wild-type and Plin1-knockout mice were dissected and fixed in 10% neutral buffered formalin (Biosesang, FR2013-000-00) overnight. Paraffin-embedded adipose tissues were sliced on a microtome (Leica Biosystems, RM2245) into 4 μm sections and placed on slide glasses. All sections were processed carefully under the same conditions. The sections were deparaffinized as follows: 10 min in xylene two times, 3 min in 100% ethanol three times, 2 min in 90% ethanol, 2 min in 80% ethanol, and 2 min in 70% ethanol. The slides were pre-incubated with 3% $H_2O_2$ (DUKSAN GENERAL SCIENCE, 3059) in distilled water for 20 min, boiled for 3 min in citrate buffer (pH 6.0, Sigma-Aldrich, C9999) for antigen retrieval, and cooled to room temperature. The blocking was performed with 2.5% normal goat serum (Vector Labs, 30024) for 1 h at room temperature.

For immunohistochemistry staining, the slides were treated with primary antibody against F4/80 (Cell Signaling, 70076, 1:200) overnight at 4 °C. Subsequently, the slides were washed three times and incubated with ImmPRESS-HRP horse anti-rabbit IgG polymer reagent (Vector Labs, 30026) for 1 h at room temperature. ImmPACT DAB Peroxidase Substrate Kit (Vector Labs, SK-4105) was used to develop slides. Tissues were counterstained with hematoxylin. The images were obtained using Pannoramic Scanner (3D HISTECH) and captured by CaseViewer software (3D HISTECH).

For immunofluorescence staining, the slides were incubated at 4 °C overnight with anti-Perilipin 1 (Invitrogen, PA1-1051, 1:500). Subsequently, the slides were washed three times and incubated with goat anti-rabbit IgG cross-adsorbed secondary antibody, Alexa Fluor 647 (Invitrogen, A21245, 1:1,000), for 1 h at room temperature. All slides were counterstained with DAPI (ImmunoBioScience, AR-6501-01), imaged using a Zeiss confocal microscope (Carl Zeiss, LSM800), and analyzed with Zen software (version 2.6)[48].

## Hematoxylin and eosin (H&E) staining

For H&E staining, tissue sections were deparaffinized with xylene and rehydrated by ethanol gradient from 100% to 70%. The slides were stained with filtered Harris hematoxylin (Thermo Fisher Scientific, 6765007) and eosin (BBC Biochemical, 3610). The excess dye was washed with 1% HCl (DUKSAN GENERAL SCIENCE, 1129). The slides were dehydrated with serial ethanol, cleaned with xylene, and mounted with a mounting medium (Thermo Fisher Scientific, 4111). The images were obtained using Pannoramic Scanner (3D HISTECH) and captured by CaseViewer software (3D HISTECH)[48]

## Statistics and reproducibility

Statistical analysis was performed using GraphPad Prism. Data are presented as the mean ± standard deviation from four independent experiments in cell culture experiments. In the case of the embryonic experiment, the significance was assessed using an unpaired Student's $t$ test. No statistical method was used to predetermine the sample size. No data was excluded from the analysis. The experiments were not randomized. The Investigators were not blinded to allocation during experiments and outcome assessment.

## Reporting summary

Further information on research design is available in the Nature Portfolio Reporting Summary linked to this article.

## Data availability

The NGS data used in this study have been deposited in the NCBI Sequence Read Archive (SRA) under the accession number PRJNA1208416. All data supporting the conclusions of this study are included in the main manuscript, Supplementary Information, and Supplementary Data sections. Source data are provided in this paper.

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

## Acknowledgements

This study was supported by the Chung Yang, Cha Young Sun, & Jang Hi Joo Memorial Fund. This work was supported by the National Research Foundation of Korea (NRF) grant funded by the Korea government (MSIT) (Korea Mouse Phenotyping Project, NRF-2014M3A9D5A01074636, NRF-2014M3A9D5A01075128, RS-2023-00261905, RS-2023-00220894, RS-2023-NR077033, and RS-2024-00441068). And the illustrations were generated using BioRender.com.

## Author contributions

T.Y.J., D.E.Y., S.P.K., J.Y., H.L., D.L., J.K.S., and K.K. designed the study. T.Y.J., D.E.Y., S.P.K., J.Y., S.-Y.L., S.O., S.J., J.E.P., S.B.L., S.-J.P., S.K., S.K.K., S.E.L., and H.S.K. performed the experiments. J.K.S. and K.K. supervised the research. All authors discussed the results and commented on the manuscript.

## Competing interests

T.Y.J., D.E.Y., J.K.S., and K.K. have filed a patent application with the Korea Patent Office (Application No. 10-2024-0046182) related to the generation of genetically engineered animals using the CRISPR-VLP system and its applications described in this study. The other authors have no competing interests to declare.

## Additional information

[1]Department of Physiology, Korea University College of Medicine, Seoul, Republic of Korea. [2]Department of Biomedical Sciences, Korea University College of Medicine, Seoul, Republic of Korea. [3]Laboratory for Genomic and Epigenomic Medicine, Research Institute for Veterinary Science, and BK21 PLUS Program for Creative Veterinary Science Research, College of Veterinary Medicine, Seoul National University, Seoul, Republic of Korea. [4]Transgenic core facility, Max-Planck Institute of Biochemistry, Martinsried, Germany. [5]Korea Model animal Priority Center, Seoul National University, Seoul, Republic of Korea. [6]Laboratory of Developmental Biology and Genomics, Research Institute for Veterinary Science, and BK21 PLUS Program for Creative Veterinary Science Research, College of Veterinary Medicine, Seoul National University, Seoul, Republic of Korea. [7]Department of Convergence Medicine, Korea University College of Medicine, Seoul, Republic of Korea. [8]Department of Life Science, Ewha Womans University, Seoul, Republic of Korea. [9]Research Animal Resource Center, Korea Institute of Science and Technology (KIST), Seoul, Republic of Korea. [10]Department of Anatomy, Korea University College of Medicine, Seoul, Republic of Korea. [11]Interdisciplinary Program in Bioinformatics and BIO MAX/N-Bio Institute, Seoul National University, Seoul, Republic of Korea. [12]Interdisciplinary Program of Cancer Biology, Seoul National University Cancer Research Institute, Seoul, Republic of Korea. [13]These authors contributed equally: Tae Yeong Jeong, Da Eun Yoon, Sol Pin Kim, Jiyun Yang. ✉e-mail: snumouse@snu.ac.kr; kyoungmi_kim@snu.ac.kr

