## [Transparent Peer Review file · Nature Communications]

An innovative approach using CRISPR-ribonucleoprotein packaged in virus-like particles to generate genetically engineered mouse models

Corresponding Author: Professor Kyoungmi Kim

Version 0:

Reviewer comments:

Reviewer #1

(Remarks to the Author)

This is an interesting paper describing the use of virus-like particles (VLPs) for generating transgenic mice. Two observations are new: 1) mouse embryo zona pellucida is permeable to LV or VLPs; 2) VLP-delivered Cas9 RNPs can efficiently generate gene modified mice. However, the following concerns need to be addressed.

Major concerns:

- 1) This paper used 10% or 20% VLPs to generate gene modified mice. I could not find information on the concentration of the particles used. The authors mentioned P30 ELISA for VLP quantification, but I could not find the quantification results. Without VLP concentrations, 10% or 20% VLPs did not provide much useful information. The authors are suggested to provide this information to help the readers understand how much particles were used. This issue applies to all figures. In some cases, volumes were used, in others, % volume was used. These need to be changed to something that reflects the quantity of the particles used.
- 2) The authors used PEG to purify the VLPs. Using ELISA to quantify PEG purified VLPs will greatly overestimate the concentration, possibly due to the precipitation of free P30 in the supernatant by PEG. The authors are suggested to quantify P30 by ELISA before PEG purification, with kits that can omit the free p30 from the medium.
- 3) The authors can do a better job to emphasize the observation of permeability of the zona pellucida to LV or VLPs. AAV, BHV1 and nanoparticles have been shown to penetrate ZP. If I am correct, directly using LV or VLPs to penetrate the ZP for mouse genome modification has not been tested.
- 4) Fig.1 e-i, Fig.2b, Fig.3 b-c, Fig. 4 e-f, the authors should list the percentages of embryos from each group with editing rates of <0.5% (detection limit of NGS), <5%, 5%-10%, 10%-20%, 30%-40% and so on.
- 5) Supplementary Fig.5C, it seemed that the authors 5 F1 carriers from the breeding of the chimeric founder. This information should be provided in the main text, rather than letting the readers to look for it.
- 6) Fig.5g, the authors should indicate the percentage of integration positive embryos.
- 7) Fig.6, the authors used the combi method for multiple gene editing. The authors should emphasize that Cas9 cannot be "combi"-ed with ABE. Fig.6 b and c, e and f, can be combined for a better comparison between combi and double VLPs.
- 8) In Fig.6, the authors used numbers of VLPs. How numbers of VLPs were calculated? Why not using this unit in other figures?

Minor concerns:

- 1) For reference 34, the authors cited a correction, will it be better to cite the original publication?

Reviewer #2

(Remarks to the Author)

In this manuscript, Jeong TY, Yoon DE, Kim SP, Yang J et al., describe a simple technical approach to generate genetically modified mice using virus-like particles (VLPs) that deliver CRISPR ribonucleoprotein complexes. The methodology, dubbed CRISPR-VIM (for CRISPR-VLP-induced targeted mutagenesis), relies on the fusion spCas9 or base editors to the Gag structural protein of Murine Leukemia Virus in order to encapsidate them within pseudo-viral particles. These VLPs are directly incubated with mouse zygotes or during in vitro fertilization, and are able to pass through the zona pellucida and transduce the embryo to deliver their protein/sgRNA payload.

Using CRISPR-VIM the authors are able to generate Knock-out (with spCas9), perform targeted mutagenesis (with different

Base editors), or Knock-in (using spCas9 and AAV donor templates), cells and mice. Finally, authors used CRISPR-VIM to perform targeted editing at multiple loci in cell lines and mouse embryos.

Overall the manuscript is of good technical quality, very well written and presents a simple and inexpensive approach to genetically modify embryos without the use of microinjection or electroporation of RNPs or mRNAs. However, the technical approach is not entirely novel at the conceptual level. VLPs have been extensively used to deliver CRISPR ribonucleoproteins (including base editors) in primary cells and in vivo, as well as to generate genetically modified mice through microinjection of VLPs in the perivitelline space. Furthermore, there are already published studies showing that viral particles can diffuse through the zona pellucida and reach the zygote. Taking this into consideration, the protocol although simple and inexpensive is not as efficient as other described protocols, such as RNP electroporation (for example PMID: 29323173), that do not require overly expensive instruments or tedious microinjection either, and allow to perform Knock-out and Knock-in transgenic mice. Furthermore, it does not present any novel optimization of the VLP platform to improve its efficiency in mediating genome-editing in embryos, notably for HDR. This efficiency problem could represent a drawback when working with species that have a long breeding cycle and therefore limit the application of CRISPR-VIM to a restricted set of species. Finally, for some of the most interesting approaches (such as Knock-In and knock-out of multiple genes in a single individual), the authors do not show whether these genetic modifications can be efficiently transmitted to the next generation.

Nevertheless, one advantage of CRISPR-VIM could reside in its versatility to use Cas9 variants that are difficult to produce as recombinant proteins, such as base editors (as tested by authors) or prime-editors. Furthermore, it could be an interesting approach to perform genome editing in embryos of species that do not tolerate electroporation, but this was not explored by authors.

Below you will find some major and minor points that, in my opinion, should be addressed by authors in order to improve the manuscript.

Major-points:

- Knock-in experiments shown in Figure 5 lack quantitative assessment in mouse embryos. Authors should also test how efficient is insertion of longer sequences (i.e. append a GFP sequence to an endogenous coding sequence or insert a large protein tag such as Halo-tag). Furthermore, authors should show that the knocked-in sequences can be effectively transmitted to the next generation.

- If I understood correctly, figure 6i shows the percentage of embryos displaying no detected mutations, one, two or three mutations. However, the fact that each mutation is found within a given embryo does not necessarily mean that they occurred in the same cells. It is therefore essential for authors to test how efficiently the double or triple mutant alleles can be transmitted to the offspring. This needs to be quantitatively assessed so that readers that are potentially interested in using the CRISPR-VIM approach to generate, in a single-step, a stable strain with multiple mutated loci know what they can realistically expect in terms of efficiency. Will they have to screen tens or hundreds of F2 mice to identify those that simultaneously bear all three mutant alleles?

Minor points:

- Figure 3e and h. Authors should show pictures of the F0 individuals so that readers can assess the degree of mosaicism.

- Supplementary Figure 1. There is a mistake in the figure legend. Panels (d-f) correspond to A-to-G conversion efficiencies and not to indels. Similarly, panels (g-i) correspond to indels and not to A-to-G conversion efficiencies.

- Figure 1b: Could the authors write the exact indel % for each targeted locus instead of only showing the color code?

- How come the individual in Fig. 3f, which displays a 42% A-to-G conversion does not appear in the Fig. 3e plot (which tops at 30% editing efficiency)?

- Low efficiency of the AncBE4max approach in mice (Figure 4). Could the authors show whether codon optimization improves the amount of AncBE4max that is loaded within VLPs compared to the WT AncBE4max sequence?

Version 1:

Reviewer comments:

Reviewer #1

(Remarks to the Author)

The authors have adequately addressed my concerns.

Reviewer #2

(Remarks to the Author)

The authors have satisfactorily answered to all of the points raised during the first round of review. The new datasets indicating transmission of the desired edits to the F1 and F2 generations (including those of multiple editing effects) have

significantly improved the manuscript. This revised version will be very useful for potential users of the CRISPR-VIM to have a realistic expectation of what they can achieve with the method and also know where there is still room for improvement.

Below you will find a list of minor edits that could be introduced in the text:

Lines 82-83: The text mentions that gene editing reaches saturation in groups treated with 10 or 25 μ l of VLPs, however this seems to be the case only for ABE8e conditions. SpCas9 samples seem to reach saturation starting from 5 μ l for most genes. I advice changing the sentence to : "However, for most targeted genes, gene editing efficiency reached saturation in groups treated with 5, 10 or 25 μ l of VLPs".

Lines 86-88: The current text mentions editing efficiencies of up to 93.8% for SpCas9 VLPs and 98.4% for ABE8e VLPs. However, Figure 1b shows editing efficiency of up to 99.7% for SpCas9 and Figure 1c shows conversion efficiencies of up to 98.2%. Could the authors correct the values mentioned in the text or clarify whether they are talking about the maximum editing efficiency observed in a single replicate (Figures 1b and 1c only show the mean values of the 4 replicates). In any case, the maximum editing efficiency for the SpCas9 has to be corrected because the mean value displayed in the figure is higher than the one mentioned in the text.

Lines 95-98: Supplementary Fig. 2 indicates a gene editing efficiency close to 100% in some of the embryos tested. Since authors disclosed the efficiency of A to G conversion in the following figure panel, I think it could be worth mentioning the SpCas9 efficiency as well in the text.

Lines 130-132: Was the mouse used in this experiment (Fig 2f, g) an F2 homozygous Plin1-knockout mouse obtained by crossing two of the Plin1-knockout F1 mice? If so, could the authors mention it in the text?

Supplementary Figure 7 legend: Could the authors change "infection" to "transduction" when referring to experiments with AAV6 and DJ serotypes?

Supplementary Figure 8c and d: Could authors use the same range in y-axis values for the "Desired edit (%)" and "Indel (%)"? It will ease the visual comparison between the two classes of edits.

Line 275: I think that "groundbreaking" should be replaced by "novel".

Line 291: Could authors change "infectivity" by "transduction efficiency"?

Point-by-point Responses

Thank you for your constructive feedback, which has significantly improved our manuscript. In response to the reviewers' comments, we conducted additional experiments to evaluate the knock-in efficiency of longer sequences (e.g., 728 bp at the *Kcnq4* locus) and successfully confirmed knock-in in both embryos and offspring. Notably, we demonstrated that the knock-in genotype was successfully transmitted to the next generation. These results highlight the reliability of the CRISPR-VIM method for generating stable knock-in mouse lines. Furthermore, as suggested, we established an additional mouse line with inheritable multi mutations. We also addressed other major and minor concerns, including providing detailed VLP quantification, clarifying legends, and improving figure clarity to strengthen the manuscript.

REVIEWER COMMENTS

Reviewer #1:

This is an interesting paper describing the use of virus -like particles (VLPs) for generating transgenic mice. Two observations are new: 1) mouse embryo zona pellucida is permeable to LV or VLPs; 2) VLP-delivered Cas9 RNPs can efficiently generate gene modified mice. However, the following concerns need to be addressed.

Response: Thank you for taking the time to review our manuscript and for your thoughtful feedback. We have added additional data and clarifications below to address your concerns. Your thoughtful feedback has been invaluable in enhancing the clarity and impact of our findings.

Major concerns:

1) This paper used 10% or 20% VLPs to generate gene modified mice. I could not find information on

the concentration of the particles used. The authors mentioned P30 ELISA for VLP quantification, but I could not find the quantification results. Without VLP concentrations, 10% or 20% VLPs did not provide much useful information. The authors are suggested to provide this information to help the readers understand how much particles were used. This issue applies to all figures. In some cases, volumes were used, in others, % volume was used. These need to be changed to something that reflects the quantity of the particles used.

Response: We agree with your suggestion. To determine the exact particle concentration, we quantified VLPs using P30 ELISA for all targets in the experiments. Based on these results, we have included the exact particle concentrations in Supplementary Fig. 10 and updated each figure legend accordingly. This additional information helps readers understand the specific quantities of particles used in the experiments, providing greater clarity and consistency.

2) The authors used PEG to purify the VLPs. Using ELISA to quantify PEG purified VLPs will greatly overestimate the concentration, possibly due to the precipitation of free P30 in the supernatant by PEG. The authors are suggested to quantify P30 by ELISA before PEG purification, with kits that can omit the free p30 from the medium.

Response: Thank you very much for your insightful comment. We apologize for not clearly mentioning in the original manuscript that our calculations indicated only 20% of the observed values from the ELISA assay correspond to intact VLPs. As highlighted in the study by Renner et al. (2020), $23.8\% \pm 2.2$ of the p30 capsid protein is associated with intact viral particles, while the remaining portion is likely fragmented or free in the medium. Similarly, the foundational study by Banskota et al. (2022) on engineered VLPs also estimated that a minimum of 20% of the detected p30 is related to intact viral particles. Based on this evidence, we adopted this 20% value as a baseline for calculating VLP titers in our study.

All VLP counts reported in our research were calculated with the assumption that each VLP contains 1800 molecules of p30 and that 20% of the p30 is associated with intact VLP. We have now clarified this information in the “Titration of VLP-packaged CRISPR-RNP” section of the

methods, specifically on page 16, line 337-338.

Additionally, we attempted to identify and utilize ELISA kits that could omit free p30 from the medium, as suggested. However, we could not find an appropriate method to address this issue at this time. We will continue exploring possible solutions to address your valuable suggestions in future studies.

3) The authors can do a better job to emphasize the observation of permeability of the zona pellucida to LV or VLPs. AAV, BHV1 and nanoparticles have been shown to penetrate ZP. If I am correct, directly using LV or VLPs to penetrate the ZP for mouse genome modification has not been tested.

Response: Thank you for your suggestion. In the previous figure, we did not provide direct evidence of VLP penetration through the ZP. Based on your recommendation, we conducted additional experiments by adding CRISPR-RNPs alone or VLPs containing CRISPR-RNPs to the zygote culture medium and cultured them together. We then compared gene editing efficiencies at the morula and blastocyst stages, with the results now included in Supplementary Fig. 2. The data shows no gene editing in the CRISPR-RNPs-alone group, while VLPs packaged with CRISPR-RNPs demonstrated high editing efficiencies of up to 100%. These results strongly confirm that VLPs can penetrate the ZP and deliver CRISPR-RNPs into the cytoplasm of the zygote. Subsequently, the CRISPR-RNPs are transported into the nucleus, where they efficiently edit the target DNA. This emphasizes the potential of VLPs as a robust and non-invasive tool for genome modification in embryos.

4) Fig.1 e-i, Fig.2b, Fig.3 b-c, Fig. 4 e-f, the authors should list the percentages of embryos from each group with editing rates of <0.5% (detection limit of NGS), <5%, 5%-10%, 10%-20%, 30%-40% and so on.

Response: Thank you for your suggestion. We followed your advice and grouped the results into different efficiency intervals (<1 %, 1~5 %, 5~10 %, 10~20 %, 20~30 %, 30~40 %, and >40%).

Across all three targets, we observed that 10% VLPs had a higher distribution in the lower efficiency segments, while 20% VLPs dominated the higher efficiency segments. These data have been added to Supplementary Fig. 6.

5) Supplementary Fig.5C, it seemed that the authors 5 F1 carriers from the breeding of the chimeric founder. This information should be provided in the main text, rather than letting the readers to look for it.

Response: Thank you for your suggestion. We have reorganized our figures to make them easier to find. In response to your comments, we have moved Supplementary Fig. 5a-b to main Fig. 2d-e, and Supplementary Fig. 5c-d to main Fig. 3h-i.

6) Fig.5g, the authors should indicate the percentage of integration positive embryos.

Response: We agree with your feedback. To provide a more accurate assessment of knock-in efficiency, we conducted an experiment to create a knock-in mouse model using *Kcnq4* (728 bp). In this experiment, we observed a successful knock-in rate of 25% (2 out of 8 embryos). When applying this method to generate mouse models, we found that 8.3% of offspring (1 out of 12) carried the intended knock-in genotype under the same conditions. Additionally, we confirmed that the designed knock-in genotype was successfully transmitted to the next generation. These results highlight the potential of this method for creating stable knock-in mouse lines. All relevant data have been included in Fig. 5, with detailed descriptions on page 10, lines 215-225.

7) Fig.6, the authors used the combi method for multiple gene editing. The authors should emphasize that Cas9 cannot be “combi”-ed with ABE. Fig.6 b and c, e and f, can be combined for a better comparison between combi and double VLPs.

Response: Thank you for your suggestion. We emphasized that Cas9 cannot be “combi”-ed with

ABE and added the sentence on page11, lines 231-234: “However, this approach does not allow the combined use of Cas9 and ABE, as neither tool can selectively distinguish between individual sgRNAs, which may result in unintended mutations.” Additionally, based on your comments, we combined the figures (Fig. 6b, d) to facilitate better comparisons.

8) In Fig.6, the authors used numbers of VLPs. How numbers of VLPs were calculated? Why not using this unit in other figures?

Response: Thank you for your suggestion. The number of VLPs was calculated using the MuLV Core Antigen ELISA kit (Cell Biolabs; VPK-156), following the methods described in Renner et al. (2020) and Banskota et al. (2022). We assumed that 20% of the observed p30 was associated with VLPs and that each VLP contained 1800 molecules of p30. Details of this calculation are included in the 'Titration of VLP-packaged CRISPR-RNP' section in the methods on page 16. To address your concern, we have additionally included VLP numbers in the figure legends to enhance clarity.

Minor concerns:

1) For reference 34, the authors cited a correction, will it be better to cite the original publication?

Response: Thank you for your suggestion. We prefer to use original publications and have updated reference 34 accordingly.

Reviewer #2:

In this manuscript, Jeong TY, Yoon DE, Kim SP, Yang J et al., describe a simple technical approach to generate genetically modified mice using virus-like particles (VLPs) that deliver CRISPR ribonucleoprotein complexes. The methodology, dubbed CRISPR-VIM (for CRISPR-VLP-induced targeted mutagenesis), relies on the fusion spCas9 or base editors to the Gag structural protein of Murine Leukemia Virus in order to encapsidate them within pseudo-viral particles. These VLPs are directly incubated with mouse zygotes or during in vitro fertilization, and are able to pass through the zona pellucida and transduce the embryo to deliver their protein/sgRNA payload.

Using CRISPR-VIM the authors are able to generate Knock-out (with spCas9), perform targeted mutagenesis (with different Base editors), or Knock-in (using spCas9 and AAV donor templates), cells and mice. Finally, authors used CRISPR-VIM to perform targeted editing at multiple loci in cell lines and mouse embryos.

Overall the manuscript is of good technical quality, very well written and presents a simple and inexpensive approach to genetically modify embryos without the use of microinjection or electroporation of RNPs or mRNAs. However, the technical approach is not entirely novel at the conceptual level. VLPs have been extensively used to deliver CRISPR ribonucleoproteins (including base editors) in primary cells and in vivo, as well as to generate genetically modified mice through microinjection of VLPs in the perivitelline space. Furthermore, there are already published studies showing that viral particles can diffuse through the zona pellucida and reach the zygote. Taking this into consideration, the protocol although simple and inexpensive is not as efficient as other described protocols, such as RNP electroporation (for example PMID: 29323173), that do not require overly expensive instruments or tedious microinjection either and allow to perform Knock-out and Knock-in transgenic mice. Furthermore, it does not present any novel optimization of the VLP platform to improve its efficiency in mediating genome-editing in embryos, notably for HDR. This efficiency problem could represent a drawback when working with species that have a long breeding cycle and therefore limit the application of CRISPR-VIM to a restricted set of species. Finally, for some of the most interesting approaches (such as Knock-In and knock-out of multiple genes in a single individual), the authors do not show whether

these genetic modifications can be efficiently transmitted to the next generation.

Nevertheless, one advantage of CRISPR-VIM could reside in its versatility to use Cas9 variants that are difficult to produce as recombinant proteins, such as base editors (as tested by authors) or prime-editors. Furthermore, it could be an interesting approach to perform genome editing in embryos of species that do not tolerate electroporation, but this was not explored by authors.

Below you will find some major and minor points that, in my opinion, should be addressed by authors in order to improve the manuscript.

Response: Thank you so much for your detailed review of our paper. As you mentioned, while this study may not be technically novel, it represents a significant milestone as the first application of VLPs in generating mutant mouse lines without requiring microinjector or electroporator. Developing a method that enables any laboratory to efficiently create animal models with specific mutations holds immense potential to accelerate various therapeutic investigations. Prior efforts, such as nanoblades (Mangeot et al., 2019), although promising, still relied on microinjection techniques, which require highly skilled researchers and expensive equipment, thereby limiting accessibility to many laboratories.

Furthermore, while RNP electroporation remains an efficient method for gene editing delivery, it faces challenges in purifying proteins for diverse CRISPR systems, such as base editor or prime editor. Our study offers an alternative approach to overcome these limitations. Additionally, exploring genome editing in embryos of species intolerant to electroporation presents intriguing possibilities. However, given the significant differences in embryo characteristics across species, each case will require independent exploration in the future. This has been noted in the discussion section on page 14, lines 300-304: “This method has the potential to be applied not only to mice but also to medium- and large-sized animals. However, further research is required to validate its applicability to medium- and large-sized animals. Therefore, the CRISPR-VIM method presents various possibilities as a potential method for embryonic gene editing and the creation of animal models.”

Below, we have provided point-by-point responses addressing the comments outlined in the

review.

Major-points:

- Knock-in experiments shown in Figure 5 lack quantitative assessment in mouse embryos. Authors should also test how efficient is insertion of longer sequences (i.e. append a GFP sequence to an endogenous coding sequence or insert a large protein tag such as Halo-tag). Furthermore, authors should show that the knocked-in sequences can be effectively transmitted to the next generation.

Response: We agree with your feedback. To address this, we first performed a knock-in test using *Kcnq4* (728 bp) insertion donor to evaluate the insertion efficiency of longer sequences, such as GFP-length fragment. In the experiment, successful knock-in was confirmed in 2 out of 8 embryos. Using this method to generate mouse models, we observed that 1 out of 12 offspring exhibited the desired knock-in genotype. Furthermore, we verified that the designed knock-in genotype was successfully transmitted to the next generation. All related data have been added to Fig. 5, and the details are described on page 10, lines 215-225.

- If I understood correctly, figure 6i shows the percentage of embryos displaying no detected mutations, one, two or three mutations. However, the fact that each mutation is found within a given embryo does not necessarily mean that they occurred in the same cells. It is therefore essential for authors to test how efficiently the double or triple mutant alleles can be transmitted to the offspring. This needs to be quantitatively assessed so that readers that are potentially interested in using the CRISPR-VIM approach to generate, in a single-step, a stable strain with multiple mutated loci know what they can realistically expect in terms of efficiency. Will they have to screen tens or hundreds of F2 mice to identify those that simultaneously bear all three mutant alleles?

Response: Thank you for your comments. We created mouse models by applying multi-target gene editing using CRISPR-RNP packaged in VLPs, employing both combi and triple approaches. Among the 82 mice born through the combi approach, 6 had mutations in both

targets, and 24 had mutations in only one of the targets. In the triple approach, out of 59 mice, none carried mutations in all three targets, but 11 had mutations in one target, and 1 had mutations in two targets. These results suggest that while multi-target editing up to four targets is feasible in cells, limiting the number of targets to two is more efficient when working with mouse embryos.

To evaluate the efficiency of transmitting mutant alleles to offspring, we used mice born from combi gene editing. In F0 mice with two mutations, if each target had a mutation rate exceeding 10%, the mutation was predominantly transmitted to the F1 generation. Additionally, by crossbreeding heterozygous F1 mice carrying both mutations, we obtained 17 F2 mice, one of which was homozygous for both mutations. These findings confirm that multi-target mutations are inherited according to Mendelian ratios, enabling traditional breeding methods to develop stable strains with multiple mutated loci. We have noted this on page 11-12, lines 246-263.

Minor points:

- Figure 3e and h. Authors should show pictures of the F0 individuals so that readers can assess the degree of mosaicism.

Response: Thank you for your suggestion. We have attached a picture of a *Tyr* H420R heterozygous mouse (left) and a wild-type (WT) mouse (right). It is important to note that for the *Tyr* gene, the phenotype appears only when the H420R mutation is homozygous. Therefore, in the *Tyr* mutant F0, no visible coat color change is observed, despite achieving an editing efficiency of 42.4%.

- Supplementary Figure 1. There is a mistake in the figure legend. Panels (d-f) correspond to A-to-G conversion efficiencies and not to indels. Similarly, panels (g-i) correspond to indels and not to A-to-G conversion efficiencies.

Response: Thank you for bringing this to our attention. We have corrected the text as follows:
" a-c, g-i, Indel frequencies at *TTN*, *HEK3*, and *HBB* targets in (a-c) HEK293T or (g-i) ARPE19 cells, based on the treatment volume of SpCas9/sgRNA packaged in VLPs. d-f, j-l, A-to-G conversion efficiencies using ABE8e/sgRNA packaged in VLPs in two human cell lines: (d-f) HEK293T or (j-l) ARPE19 cells (n = 4 for all conditions)." This correction has been made in the Supplementary Fig. 1 legend on page 4, lines 28-32.

- Figure 1b: Could the authors write the exact indel % for each targeted locus instead of only showing the color code?

Response: Thank you for your suggestion. We have added the exact gene editing percentages to the heat maps in Fig. 1b, 1c, Fig. 4b, and 4d. To ensure reliability, the experiment was conducted three times, and the displayed values now represent the averages .

- How come the individual in Fig. 3f, which displays a 42% A-to-G conversion does not appear in the Fig. 3e plot (which tops at 30% editing efficiency)?

Response: Thank you for pointing that out. We apologize for the unclear description. Fig. 3e depicts the editing outcomes of morula and blastocysts cultured in vitro after VLP co-culturing. In contrast, Fig. 3f illustrates data obtained from mouse pups that were transferred to a foster mother at the 2-cell stage following VLP treatment. A distinct pipeline is delineated in Fig. 3a. To clarify, we have revised the y-axis title to "A-to-G conversion in morula & blastocyst" and updated the legends for Fig. 3e and 3f accordingly, as detailed on page 29, lines 646-649.

- Low efficiency of the AncBE4max approach in mice (Figure 4). Could the authors show whether codon optimization improves the amount of AncBE4max that is loaded within VLPs compared to the WT AncB4max sequence?

Response: CBE-based VLPs contain a FLAG tag at the Cas9 N-terminus. The quantification of Cas9 loaded within the VLP was performed using ELISA with the FLAG tag. Banskota et al. (2022) demonstrated that the approximate amount of Cas9 loaded within the VLP was 400 nM. Our findings indicate that the codon-optimized CBE (OPT) exhibited roughly twice the loading capacity compared to the wild-type CBE (WT). This suggests that codon optimization enhances the packaging efficiency of CBEs into VLPs, leading to improved editing efficiency. Quantitative data on the enhanced editing efficiency is provided in

Supplementary Fig. 5c, which demonstrates the relationship between increased CBE loading and improved genome editing outcomes.

Point-by-point Responses

REVIEWER COMMENTS

Reviewer #1:

The authors have adequately addressed my concerns.

We sincerely appreciate your time and effort in reviewing our work. Your thoughtful feedback has been invaluable in enhancing the clarity and overall quality of our manuscript. Thank you for your insightful evaluation.

Reviewer #2:

The authors have satisfactorily answered to all of the points raised during the first round of review. The new datasets indicating transmission of the desired edits to the F1 and F2 generations (including those of multiple editing effects) have significantly improved the manuscript. This revised version will be very useful for potential users of the CRISPR-VIM to have a realistic expectation of what they can achieve with the method and also know where there is still room for improvement.

We sincerely appreciate your positive feedback and thoughtful evaluation of our revised manuscript. We are glad to hear that the additional datasets have strengthened the manuscript and provided valuable insights for potential users of CRISPR-VIM.

Minor edits

Lines 82-83: The text mentions that gene editing reaches saturation in groups treated with 10 or 25 μ l of VLPs, however this seems to be the case only for ABE8e conditions. SpCas9 samples seem to reach saturation starting from 5 μ l for most genes. I advice changing the sentence to : "However, for most targeted genes, gene editing efficiency reached saturation in groups treated with 5, 10 or 25 μ l of VLPs".

Thank you for your valuable comment. We have revised the sentence in lines 84-86 according to your suggestion to more accurately reflect the data. The updated text: 'However, for most targeted genes, gene editing efficiency reached saturation in groups treated with 5, 10, or 25 μ l of VLPs.' Additionally, we have reviewed your comment and included the exact values for the single replicate. The precise numbers are also available in the source data.

Lines 86-88: The current text mentions editing efficiencies of up to 93.8% for SpCas9 VLPs and 98.4% for ABE8e VLPs. However, Figure 1b shows editing efficiency of up to 99.7% for SpCas9 and Figure 1c shows conversion efficiencies of up to 98.2%. Could the authors correct the values mentioned in the text or clarify whether they are talking about the maximum editing efficiency observed in a single replicate (Figures 1b and 1c only show the mean values of the 4 replicates). In any case, the maximum editing efficiency for the SpCas9 has to be corrected because the mean value displayed in the figure is higher than the one mentioned in the text.

Thank you for your careful review and valuable comments. We have corrected the editing efficiency values in lines 88–91 to accurately reflect the data. The revised text: 'The SpCas9/sgRNA packaged in VLPs achieved up to 99.8% gene editing efficiency on Fgfr3, Gata3, Plin1, and Tyr targets (Fig. 1b). ABE8e/sgRNA packaged in VLPs showed a high A-to-G conversion frequency of up to 98.4% in Plin1, Dnmt1, Gata3, and Kcnq4 targets (Fig. 1c).' Additionally, we have carefully considered your comments and included the exact values for the single replicate. These precise numbers are also available in the source data.

Lines 95-98: Supplementary Fig. 2 indicates a gene editing efficiency close to 100% in some of the embryos tested. Since authors disclosed the efficiency of A to G conversion in the following figure panel, I think it could be worth mentioning the SpCas9 efficiency as well in the text.

Thank you for your insightful feedback. In response to your suggestion, we have revised the text in lines 100–101 to explicitly state the SpCas9 gene editing efficiency, providing a more

detailed and comprehensive description of our findings. The updated text: 'Notably, SpCas9/sgRNA delivered via VLPs achieved gene editing efficiencies of up to 99.9% in embryos.'

Lines 130-132: Was the mouse used in this experiment (Fig 2f, g) an F2 homozygous Plin1-knockout mouse obtained by crossing two of the Plin1-knockout F1 mice? If so, could the authors mention it in the text?

Thank you for your insightful comment. Yes, the mouse used in this experiment (Fig. 2f, g) was an F2 homozygous Plin1-knockout mouse, which was generated by crossing two heterozygous F1 mice. We have revised the text in lines 136-137 to clearly state this information.

Supplementary Figure 7 legend: Could the authors change “infection” to “transduction” when referring to experiments with AAV6 and DJ serotypes?

Thank you for your valuable suggestion. As recommended, we have revised the legend for Supplementary Figure 7 by replacing 'infection' with 'transduction' when referring to experiments involving AAV6 and DJ serotypes.

Supplementary Figure 8c and d: Could authors use the same range in y-axis values for the “Desired edit (%)” and “Indel (%)”? It will ease the visual comparison between the two classes of edits.

We appreciate your careful observation and valuable suggestion. In response to your comment, we have adjusted the y-axis range in Supplementary Figure 8c and 8d to be consistent for 'Desired edit (%)' and 'Indel (%)'. This change allows for a clearer visual comparison between the two classes of edits and improves the readability of the figure.

Line 275: I think that “groundbreaking” should be replaced by “novel”.

Thank you for your keen observation and suggestion. We have replaced 'groundbreaking' with 'novel' in line 278, as recommended.

Line 291: Could authors change “infectivity” by “transduction efficiency”?

Thank you for your thoughtful suggestion. We have revised line 294 by replacing 'infectivity' with 'transduction efficiency' to ensure more precise terminology.